# Aneuploidy tolerance caused by BRG1 loss allows chromosome gains and recovery of fitness

Federica Schiavoni[1,4], Pedro Zuazua-Villar[1,4], Theodoros I. Roumeliotis[2], Graeme Benstead-Hume[2,3], Mercedes Pardo [2], Frances M. G. Pearl [3], Jyoti S. Choudhary [2] & Jessica A. Downs [1✉]

Aneuploidy results in decreased cellular fitness in many species and model systems. However, aneuploidy is commonly found in cancer cells and often correlates with aggressive growth, suggesting that the impact of aneuploidy on cellular fitness is context dependent. The BRG1 (SMARCA4) subunit of the SWI/SNF chromatin remodelling complex is frequently lost in cancer. Here, we use a chromosomally stable cell line to test the effect of BRG1 loss on the evolution of aneuploidy. BRG1 deletion leads to an initial loss of fitness in this cell line that improves over time. Notably, we find increased tolerance to aneuploidy immediately upon loss of BRG1, and the fitness recovery over time correlates with chromosome gain. These data show that BRG1 loss creates an environment where karyotype changes can be explored without a fitness penalty. At least in some genetic backgrounds, therefore, BRG1 loss can affect the progression of tumourigenesis through tolerance of aneuploidy.

---

[1] Epigenetics and Genome Stability Team, The Institute of Cancer Research, 237 Fulham Road, London SW3 6JB, UK. [2] Functional Proteomics Team, The Institute of Cancer Research, 237 Fulham Road, London SW3 6JB, UK. [3] Bioinformatics Group, School of Life Sciences, University of Sussex, Falmer, Brighton BN1 9QJ, UK. [4] These authors contributed equally: Federica Schiavoni and Pedro Zuazua-Villar. ✉email: Jessica.Downs@icr.ac.uk

Aneuploidy is a common feature of cancer cells. However, in many systems, aneuploidy is associated with a loss of fitness[1–4]. Aneuploidy can lead to cellular stress through multiple pathways[5]. First, the imbalance of proteins arising from the change in chromosome copy number can lead to the increased presence of unfolded or misfolded proteins and consequently, proteotoxic stress[6,7]. In addition, there is evidence that aneuploid cells are under metabolic stress and have elevated levels of reactive oxygen species[1,8]. Aneuploidy can also lead to increased replicative and mitotic stress, and collectively[5], these aneuploidy-associated stresses can lead to a decrease in cellular fitness.

In somatic mammalian cells, there are multiple SWI/SNF chromatin remodelling complexes, including BAF, ncBAF and PBAF[9]. One of two catalytic subunits, BRG1 or BRM (encoded by SMARCA4 and SMARCA2, respectively), is associated with each of these SWI/SNF complexes. One subunit found specifically in PBAF is the BAF180 subunit (encoded by PBRM1).

We previously found that loss of BAF180 leads to defective sister chromatid cohesion, and the depleted cells show evidence of chromosomal instability (CIN) and increased aneuploidy in U2OS cells[10]. In addition, we analysed the karyotype of the Baf180 knockout mouse embryonic stem (ES) cells and found that these had a gain of two chromosomes when compared with the parental cells[10]. There was no obvious decrease in fitness associated with the aneuploid Baf180−/− ES cells, suggesting that the increase in chromosomes was tolerated and may have provided an advantage to the cells at some point during their generation. These data suggest that SWI/SNF chromatin remodelling activity is important for preventing chromosomal instability and/or aneuploidy in mammalian cells. Genes encoding subunits of SWI/SNF, including SMARCA4 (encoding BRG1) and PBRM1 (encoding BAF180), are frequently mutated in diverse human cancers[11–13], and their contribution to regulating the cellular response to aneuploidy might be relevant to understanding their frequent loss. In order to further understand the relationship between loss of SWI/SNF and evolution of aneuploidy, here we test the impact of BRG1 loss on chromosomal stability using the HCT116 cell line, which is a near-diploid cell line with low levels of CIN[2,14]. We find that BRG1 loss leads to altered morphology, slower proliferation and an altered cell cycle profile. Over time, the BRG1 knockout cells regain fitness and this improved fitness correlates with chromosomal gain, raising the possibility that BRG1-deficient cells can tolerate aneuploidy better than BRG1-proficient cells. In support of this, we find that pathways implicated in aneuploidy tolerance are altered immediately upon loss of BRG1. Moreover, we find that BRG1-deficient cells survive better when chromosome missegregation is induced, and that surviving cells have higher rates of aneuploidy than the parental cells. These data indicate that BRG1 loss increases cellular tolerance of aneuploidy, and thus provides an environment in which cells can explore karyotypic space without the cost to cellular fitness normally associated with aneuploidy. These results have important implications for the impact of BRG1 loss on cancer progression.

## Results

### BRG1 loss in HCT116 results in an extended G1 phase, altered morphology and reduced proliferation.
To understand how loss of SWI/SNF activity impacts chromosomal instability, we used CRISPR-Cas9 methodology to create clonal cell lines in HCT116 with inactivating mutations of the SMARCA4 gene, which encodes the BRG1 subunit of the SWI/SNF complex. The HCT116 cell line was chosen because it is a near-diploid cell line that shows low levels of chromosomal instability (CIN) when compared with other cell lines and has been used previously as a model for understanding aneuploidy[2,14]. In addition, cancer-associated mutations in SMARCA4 often lead to a lack of detectable protein expression, which makes full deletion of BRG1 a relevant model for understanding the impact on tumourigenesis. We isolated 2 independent knockout (KO) clones (here referred to as 1 and 2) lacking BRG1 expression (Fig. 1a) and characterised these by sequencing (Supplementary Fig. 1a). We find that the BRG1 KO cell lines showed reduced proliferative capacity (Fig. 1b, c and Supplementary Fig. 1b) and altered morphology with more elongated cells and reduced cell-to-cell contact (Supplementary Fig. 1c). Similar results are obtained when BRG1 is depleted using siRNA in this cell line (Supplementary Fig. 1d–f), suggesting these effects are due to BRG1 loss. There was no apparent increase in cell death or senescence in the BRG1 KO cell lines, suggesting that they were moving more slowly through the cell cycle. Consistent with this idea, when cell cycle profiles of the clones were examined, we found no substantial increase in sub-G1 cells in the KO cells (Supplementary Fig. 1g). Instead, there was an increase in the proportion of G1 phase cells when compared with the parental HCT116 cell line (Fig. 1d–f), which suggests that it is this phase of the cell cycle through which they are moving more slowly.

### The fitness of the BRG1 deletion cell lines improves over time in culture.
To study the impact of BRG1 loss on chromosomal instability (CIN), we monitored cells over time. The cells were cultured over a period of 8 months and monitored at early, mid (4 months) and late (8 months) time points. Both cell lines showed improved fitness over time. Clone 1 regained BRG1 expression during the 8 months in culture (Supplementary Fig. 2a). Sequencing cells from clone 1 at late time points revealed the presence of a new 3 bp deletion that puts the coding sequence back in frame (Supplementary Fig. 2b). Clone 2 remained BRG1 negative for the duration of the time course by Western blotting (Fig. 2a), and we verified this through analysis of quantitative mass spectrometry data (Supplementary Fig. 2c). We, therefore, focused on these cells for analyses of the long-term effects of BRG1 loss on ploidy.

While the cells never reached the proliferative capacity of the parental HCT116 cells, the proliferation rate of the cell population increased at mid-time points and then further increased at late time points (Fig. 2b, c). In addition, the cells appeared to revert somewhat towards less elongated morphology with more cell contact (Supplementary Fig. 2d). When the cell cycle profile was analysed, the late time point cell population showed a very similar profile to the parental cells and no longer had a pronounced G1 phase (Fig. 2d, e and Supplementary Fig. 2e). These data suggest that the BRG1 knockout cells acquired genetic or epigenetic changes that compensate for the loss of BRG1 and allow the phenotype and behaviour of the cells to evolve towards that of the parental cell line.

### The improved fitness of the BRG1 deletion cell line correlates with a gain of chromosome 18.
To determine what changes might account for the increase in fitness in BRG1 KO clone 2, we analysed the karyotypes of the cells at different time points. We prepared mitotic spreads from samples taken at early, mid and late time points and quantified the number of chromosomes per cell. We found that there was a gain of chromosomes evident in many of the BRG1 KO cells at mid-time points (50.4%) and this increased to 80.9% at late time points (Fig. 3a, b and Supplementary Fig. 3a).

To identify the chromosomes that were gained in the BRG1 KO cells, we performed array comparative genomic hybridisation

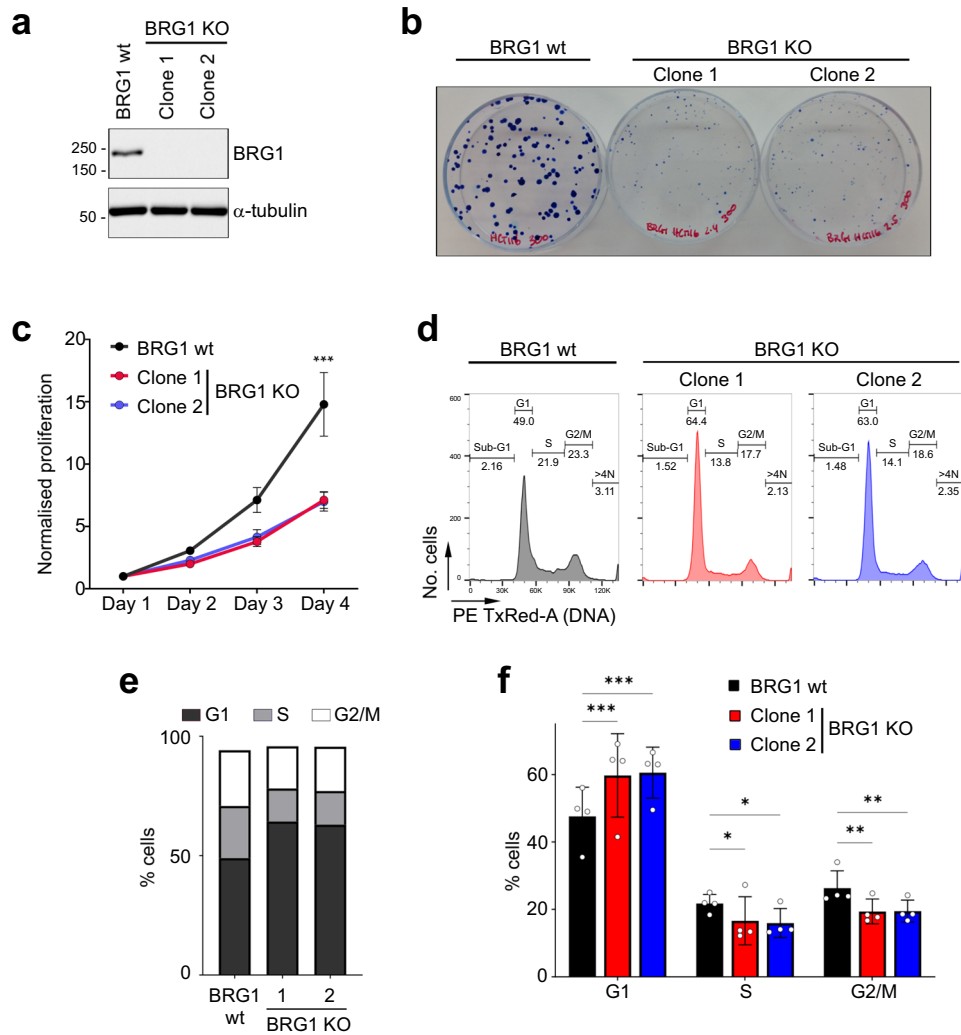

**Fig. 1 BRG1 loss leads to a loss of fitness and altered cell cycle profile. a** Western blot analysis of BRG1 in whole-cell extracts prepared from HCT116 cells and two HCT116-derived BRG1 knockout clonal cell lines. α-tubulin was used as a loading control. Similar results were obtained in three independent repeats. **b** Clonogenic assay showing the difference in colony size between HCT116 cells and two BRG1 knockout clones grown for 10 days. **c** Normalised proliferation from three independent experiments monitored by CellTiter-Glo Luminescent Cell Viability assay between HCT116 cells and two BRG1 knockout clones. Data were presented as the mean ± SEM; $n = 3$. The $p$ value was calculated with two-way ANOVA-Dunnet. ***$p < 0.001$. **d** BRG1-deficient cells have an altered cell cycle profile. DNA content was analysed by flow cytometry and representative profiles of the HCT116 or BRG1 knockout clones are shown. **e** Quantification of cells (%) in each cell cycle phase of cells analysed by flow cytometry in **d**. **f** Quantification of cells (%) in G1, S and G2/M cell cycle phases from flow cytometry profiles in HCT116 and BRG1 knockout cells. Data were presented as the mean ± SD; $n = 4$. The $p$ value was calculated with two-way ANOVA-Dunnett. *$p < 0.05$, **$p < 0.01$ and ***$p < 0.001$.

(array CGH) and found two differences in the BRG1 KO cells compared with the parental HCT116. First, duplication of chromosome 21 in the BRG1 KO cell line that is not present in the parental HCT116 cell line was present at all time points (Supplementary Fig. 3b). However, it is unlikely that this change is relevant to the phenotypes observed in the BRG1 KO cells for several reasons. First, when we analysed the metaphase spreads using fluorescence in situ hybridisation (FISH) with a probe directed against chromosome 21, we found that there was no change over time in its frequency within the population (Supplementary Fig. 3c), suggesting that it did not contribute to the change in fitness over time. In addition, the FISH analysis also showed that this was a chromosome fusion rather than an independent chromosomal gain (Supplementary Fig. 3d).

The second change identified in the CGH analysis of the BRG1 knockout cells was an additional copy of chromosome 18 (chr18) that was present at both mid and late time points (Fig. 3c). We investigated the prevalence of this chromosome gain in the population of cells at each time point using FISH with a probe directed against chr18 (Fig. 3d). We found that approximately 45% of the population had gained copies of chr18 at the mid-time point and almost all of the cells (89.7%) had more than two copies at late time points (Fig. 3e). Interestingly, we found that most of these cells had gained two copies of chr18 (Fig. 3f), and the extra copies of chr18 explain the majority of the changes in the karyotypes (Fig. 3b and Supplementary Fig. 3a). This gain of extra copies of chr18 correlates well with the gain in fitness in the population and is consistent with a model in which fitness advantages associated with the gain of chr18 allow these cells to overtake the population.

**Proteomic analysis identifies pathways that evolve in the BRG1 knockout cells.** In order to gain further insights into the changes observed in the BRG1 KO cells over time, we performed proteomics analysis using quantitative mass spectrometry. We used

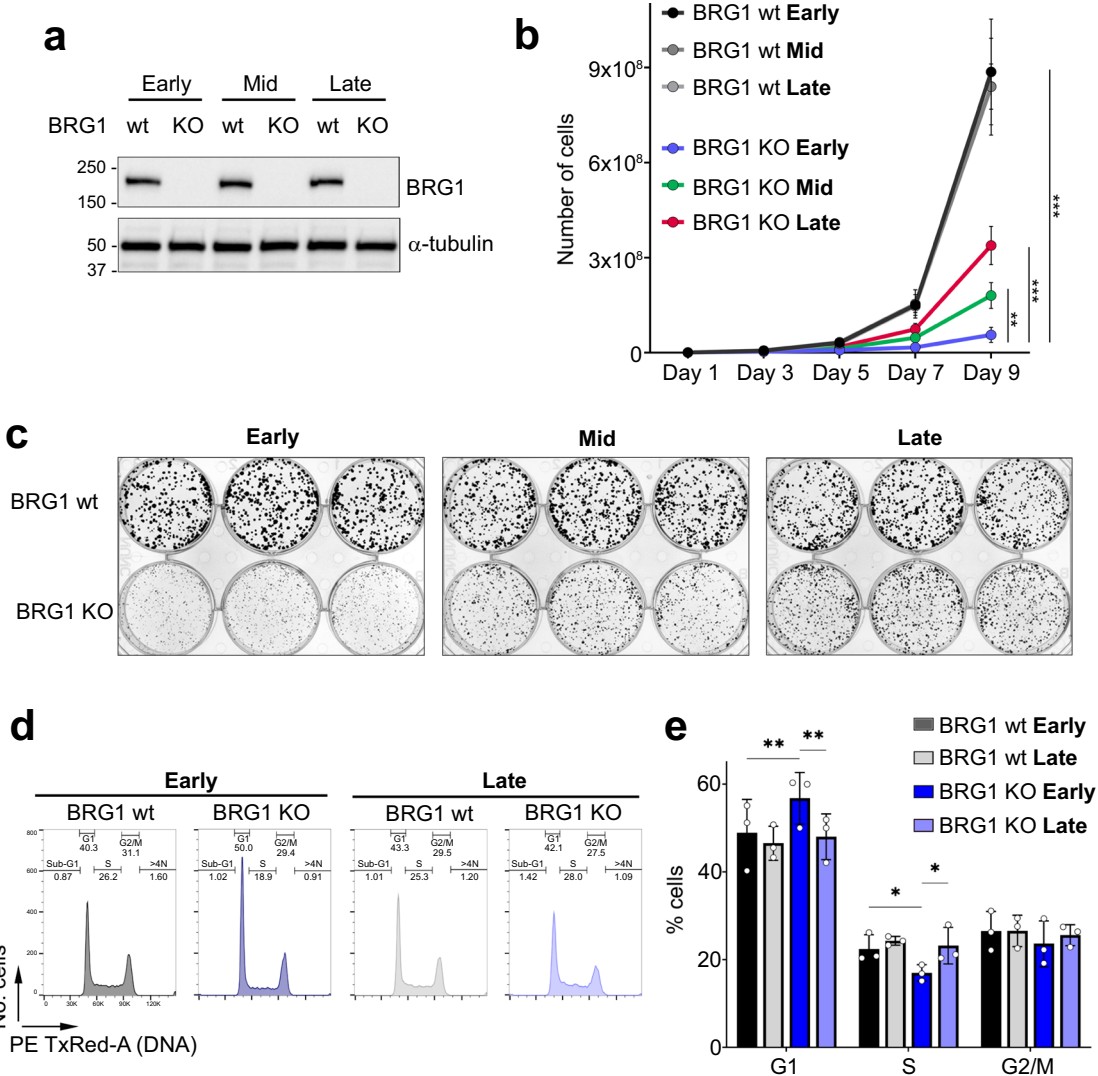

**Fig. 2 The fitness of BRG1-deficient cells improves over time. a** Western blot analysis of BRG1 in whole-cell extracts prepared from HCT116 and BRG1-deficient (KO) clone 2 cells at early, mid (4 months) and late (8 months) time points. α-tubulin was used as a loading control. Similar results were obtained in three independent repeats. **b** The proliferation rate of BRG1 knockout clone 2 improves during long-term cell culture. Cell number was monitored every 2 days over a period of 9 days in HCT116 and BRG1 KO clone 2 cells taken from early, mid (4 months) and late (8 months) time points. Data were presented as the mean ± SD; $n = 3$. The $p$ value was calculated with two-way ANOVA-Tukey. **$p < 0.01$, ***$p < 0.001$. **c** The colony size of BRG1 knockout clone 2 increases during long-term cell culture. Clonogenic assay showing the colony size of HCT116 (BRG1wt) and BRG1 knockout (KO) cells taken from early, mid (4 months) and late (8 months) time points. **d** The cell cycle profile of BRG1-deficient cells changes during long-term cell culture to a profile similar to parental HCT116 cells. Representative flow cytometry profiles in HCT116 and BRG1-deficient cells (clone 2) at early and late (8 months) time points. **e** Quantification of cells (%) in each cell cycle phase from flow cytometry profiles in HCT116 and BRG1 knockout clone 2 from early and late (8 months) time points. Data were presented as the mean ± SD; $n = 3$. The $p$ value was calculated with two-way ANOVA-Tukey. *$p < 0.05$, **$p < 0.01$.

tandem mass tag (TMT) based multiplexing to compare different cell lines. Multiplexed analysis of samples corresponding to three stages of parental and BRG1 KO clone 2 in replicates identified 8866 proteins. First, we applied principal component analysis (PCA) to determine the variation between the samples. The samples prepared at early, mid and late time points from the parental HCT116 cells clustered together (Fig. 4a), demonstrating that in comparison to the BRG1 knockout cells, the proteome of the HCT116 cell line is relatively stable over time. As expected, we found that the BRG1 knockout cells showed a considerable difference from the parental cells at the early time point (Fig. 4a). In addition, we found that the proteome of these cells clearly evolved over time (Fig. 4a).

Next, we looked at components of the SWI/SNF complex. As described above, we found no evidence of BRG1 (SMARCA4)

expression in the BRG1 knockout cells at any time point (Supplementary Figs. 2c, 3e). We also found low levels of several subunits, including BAF180 (PBRM1), BCL7B, BCL7C, BRD9 and SS18 (Supplementary Fig. 3e), likely indicating destabilization. In support of this possibility, these subunits are lost from the residual SWI/SNF complex in cell lines lacking both catalytic subunits; BRG1 and BRM (encoded by *SMARCA2*;[15]). In addition, levels of BRM were increased in the BRG1 knockout cells compared with the parental HCT116 cells (Supplementary Fig. 3e), suggesting compensatory upregulation.

We analysed the changes in the proteome of the BRG1 KO cells, both in relation to the parental HCT116 cells and over time, in order to identify molecular processes and pathways underpinning the cellular remodelling events and phenotypes (Fig. 4b). When the BRG1 KO cells at early time points are compared with

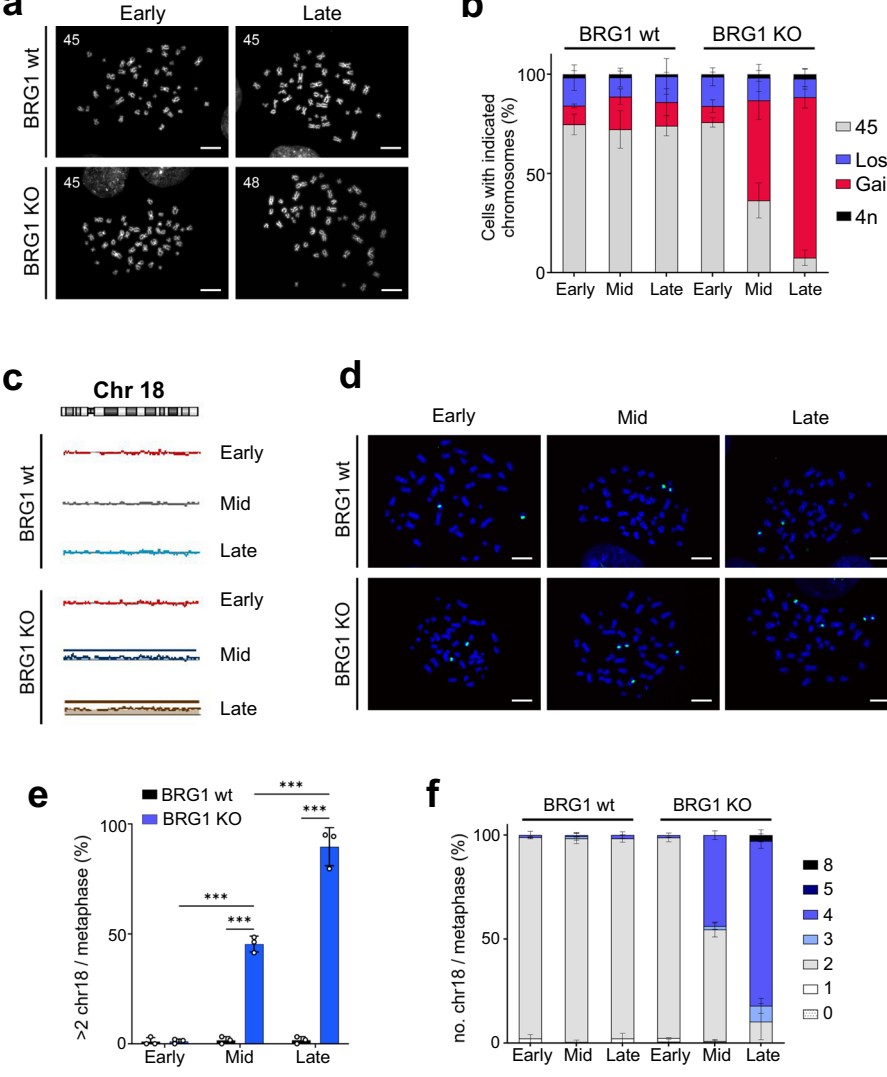

**Fig. 3 The improved fitness of BRG1-deficient cells over time correlates with the gain of additional copies of chromosome 18. a** Representative metaphase spreads prepared from HCT116 and BRG1 knockout (KO) cells (clone 2) at early and late (8 months) time points. The number of chromosomes in each metaphase are indicated. Scale bar = 10 μm. **b** The BRG1 KO cell population (clone 2), but not the HCT116, gains chromosomes during long-term cell culture. The number of chromosomes per metaphase was quantified, and the percentage of cells with gain or loss was calculated. At least 50 metaphases were analysed for each sample. Data were presented as the mean ± SD; $n = 3$. **c** BRG1 knockout cells (clone 2) acquire additional copies of chromosome 18 during long-term cell culture. Array-based comparative genomic hybridisation (array CGH) data for chromosome 18 from samples prepared from HCT116 and BRG1 knockout cells at early, mid (4 months) and late (8 months) time points. **d** Representative metaphase FISH with a probe against chromosome 18 (green) in HCT116 and BRG1 KO cells (clone 2) from early, mid (4 months) and late (8 months) time points. Scale bar = 10 μm. **e** Quantification of the number (%) of metaphase spreads with more than two chromosomes 18 copies. At least 50 metaphases were counted for each sample. Data were presented as the mean ± SD; $n = 3$. The $p$ value was calculated with two-way ANOVA-Tukey. ***$p < 0.001$. **f** There is a progressive increase in the number of copies of chromosome 18 in the BRG1 knockout cell line over time. The number of copies of chr18 per metaphase was quantified and the percentage of cells with the indicated gains are plotted. HCT116 and BRG1 KO cells from early, mid (4 months) and late (8 months) time points were analysed. Data were presented as the mean ± SD; $n = 3$.

the parental HCT116 cells, we find multiple pathways altered at high statistical significance, including metabolic pathways, hypoxia regulation and translation (Fig. 4c). The BRG1 KO cells also have downregulated pre-replicative complex activation proteins, G1/S transition components, and cell cycle-associated pathways (Fig. 4c), which reflect the altered cell cycle profile that is apparent at these early time points.

By comparing the proteome of the BRG1 KO cells from early time points with that of late time points, we can identify pathways that evolve over time. We find that some of the changes observed in the BRG1 KO cells over time bring them more in line with the proteome present in the HCT116 cells (Fig. 4d). For example, the

fatty acid metabolism pathway, which is upregulated on the loss of BRG1 (Fig. 4c), decreases at late time points (Fig. 4d) and is no longer apparent as an altered pathway when the KO cells at late time points are compared with the parental cells (Supplementary Fig. 3g). Unsurprisingly, we also identify upregulation of proteins encoded by genes on chromosome 18 (Fig. 4d and Supplementary Fig. 3f) as a signature of later time points in the BRG1 knockout cells when the cells at late time points are compared with the KO cells at early time points (Fig. 4d and Supplementary Fig. 3f) or with the parental cells (Supplementary Fig. 3g).

We find that many of the regulators of the G1/S transition are altered in the BRG1 KO cells relative to the parental HCT116

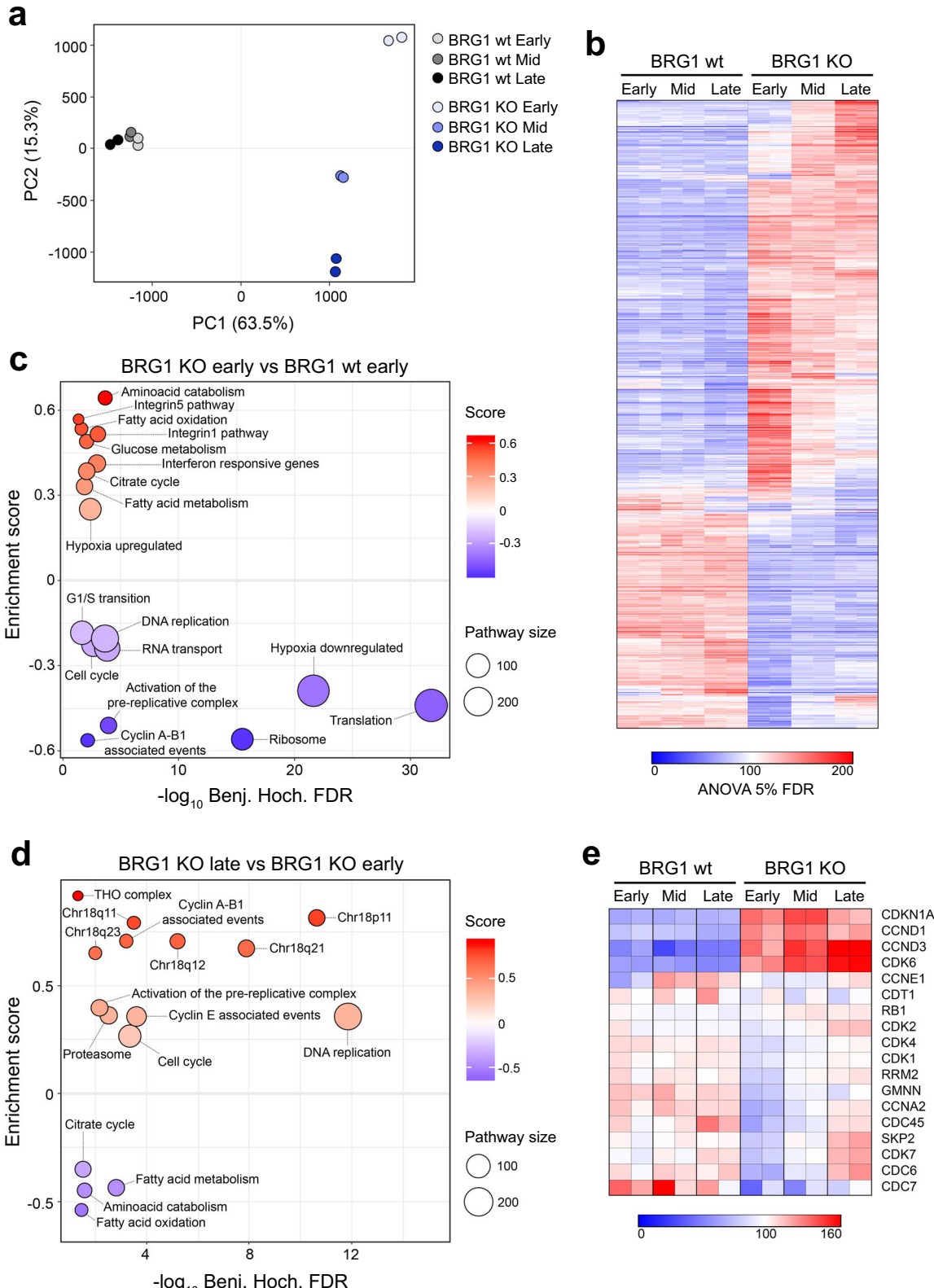

(Fig. 4e). Proteins that are normally expressed during the late G1 or S phase, such as CDC6 and CDC7, are low at early time points in the BRG1 knockout cells, which could reflect the decreased S phase population in these cells at this time point (Fig. 2d). Consistent with this interpretation, at later time points the expression levels of these proteins in the BRG1 knockout cells return to those seen in the parental HCT116 cells (Fig. 4e and

Supplementary Fig. 3g), corresponding to the change in cell cycle profile (Fig. 2d). Upregulation of proteins normally expressed in G1, such as CCND1 and CCND3, might similarly reflect the greater number of cells in the G1 phase in the BRG1 knockout population. However, levels of these proteins do not decrease at later time points when the cell cycle profile no longer shows the pronounced G1 phase population (Fig. 4e), suggesting potential

**Fig. 4 The proteome of BRG1-deficient cells shows striking differences from the parental HCT116 cells and evolves over time. a** Principal component analysis (PCA) of the proteomic profiles from HCT116 and BRG1 knockout (KO) cells (clone 2) from early, mid (4 months) and late (8 months) time points. Two 6plex proteomics experiments were performed and the dots represent biological replicates. **b** Heatmap of the proteome data using row-scaled protein abundances for the top 30% most variable proteins (based on standard deviation) with significant differential regulation (ANOVA FDR <5%). The colour key indicates the relative abundance of each protein (scale: 0–200). Duplicates are shown side by side. **c** Pathway enrichment analysis showing differentially regulated pathways in BRG1-deficient (KO) HCT116 at early time points compared with the parental HCT116 cells. Upregulated pathways (in red) and downregulated pathways (in blue) are shown according to the enrichment score. Dot size varies accordingly to the size of the pathway. **d** Pathway enrichment analysis showing changes in BRG1-deficient (KO) cells over time. For the analysis, the proteome from BRG1-deficient cells at late (8 months) time points was compared to the proteome of BRG1-deficient cells at early time points. Only pathways that are differentially regulated in the BRG1 KO cells and stable in wild-type cells over time are shown. Upregulated pathways (in red) and downregulated pathways (in blue) are shown according to the enrichment score. Dot size varies accordingly to the size of the pathway. **e** G1/S cell cycle pathway proteins are altered in BRG1 knockout cells. A scaled abundance of proteins involved in G1/S transition. HCT116 and BRG1-deficient HCT116 cells were compared at early, mid (4 months) and late (8 months) time points. The colour key indicates the relative abundance of each protein (scale: 0–160). Duplicates are shown side by side.

deregulation at posttranslational modification level associated with the activity of these proteins or alternative function of these proteins.

**BRG1 KO cells show changes associated with tolerance of aneuploidy at early time points.** We previously found that cells with depleted BAF180 had evidence of elevated CIN and that BAF180-deficient mouse cells are aneuploid[12]. In addition, BRG1-deficient mouse fibroblast cell lines display abnormal mitotic structures and replication fork progression defects, consistent with elevated CIN[16,17]. Elevated CIN would provide a mechanism by which the BRG1-deficient cells were able to explore different karyotypes and allow beneficial changes to overtake the population. We, therefore, examined the BRG1 KO cells for evidence of CIN by monitoring cellular characteristics associated with chromosome missegregation. To do this, we used both BRG1 KO clones from early time points prior to BRG1 re-expression. However, we found no substantial difference in the presence of aberrant mitotic structures or the number of micronuclei when the BRG1 KO cells were compared with the parental HCT116 (Supplementary Fig. 4). While there may be differences in CIN that are difficult to uncover due to the slow growth of the BRG1 KO cells, the evidence suggests that the levels of CIN in the absence of BRG1 in the HCT116 cell line are not significantly elevated.

Instead, we considered the possibility that loss of BRG1 led to an increased tolerance of aneuploidy. To investigate this, we examined the proteomic data to determine whether any BRG1-dependent pathway changes could have an impact on the ability of these cells to tolerate aneuploidy. The HCT116 cell line is p53 proficient, and p53 functions in response to chromosome missegregation to promote apoptosis, thereby preventing the survival of aneuploid cells[18,19]. We, therefore, looked at the p53 pathway in the BRG1 knockout cells, but while there are changes in this pathway including a small decrease in p53 levels, the pathway changes are not consistent with a loss of p53 function (Supplementary Fig. 5a). We further looked by Western blot analysis at the p53 downstream target protein p21, which is induced following p53 activation. However, rather than a decrease in p21 levels, we found that they are consistently higher in the BRG1 knockout cells at all time points (Supplementary Fig. 5b, c), suggesting that p53 is present and active in these cells, even as they become aneuploid. Overexpression of Cyclin D1 has been shown to promote tolerance of whole-genome duplication in p53-proficient cancer cells[20]. While the impact of aneuploidy might differ from that of whole-genome duplication, we found overexpression of Cyclin D1 at all time points in the BRG1 knockout cells (Fig. 4e and Supplementary Fig. 5d) and this might suggest a mechanism for overriding p53-mediated apoptosis of aneuploid cells in the BRG1 knockout cells.

We found that proteins that are known to be downregulated in response to hypoxia were downregulated in the BRG1 knockout cells at early time points (Fig. 5a), despite the cells being grown in normoxic conditions. Likewise, proteins that are upregulated in response to hypoxia are upregulated in the absence of BRG1 at early time points under normoxic growth conditions (Fig. 5b). Notably, aneuploid cells are under metabolic stress, and changes in the hypoxia-regulated HIF1α signalling pathway have been shown to promote cellular tolerance of aneuploidy[21]. HIF1α signalling also results in upregulation of glycolysis. We, therefore, looked more specifically at glycolysis and found substantial upregulation of multiple glycolysis enzymes (Fig. 5c). Moreover, two glucose transporters are markedly upregulated in the BRG1 knockout cells (Fig. 5d). These data suggest that the BRG1 knockout cells are relying more heavily on glycolysis than the HCT116 control cells, and these patterns of altered metabolic regulation are consistent with a tolerance of aneuploidy[4,21,22].

Proteotoxic stress associated with imbalances in copy number in aneuploid cells means that proteasome activity and regulation are important for tolerating aneuploidy[3,7,23–25]. The core 20 S proteasome can associate with the 19 S regulatory caps to form the 26 S proteasome, which facilitates the degradation of ubiquitin-targeted proteins. In addition, an alternative form of the proteasome exists, termed the immunoproteasome, in which three of the catalytic subunits in the 20 S complex; β1 (PSMB1), β2 (PSMB2) and β5 (PSMB5), are replaced with β1i (PSMB9), β2i (PSMB10), and β5i (PSMB8) resulting in altered proteolysis. In addition, this alternative 20 S core can associate with the PA28 regulatory complex, comprised of PSME1 and PSME2, rather than the 19 S, and can facilitate ubiquitin-independent degradation of proteins[26–28]. Notably, immunoproteasome activity has been shown to alleviate proteotoxic stress[29].

We investigated the possibility that the proteasome pathway might also be misregulated in the BRG1 deleted cells. We found that upon loss of BRG1, three proteasome subunits are strikingly upregulated (Fig. 5e). Interestingly, these are the immunoproteasome subunit PSMB9 (β1i), and PSME1 and PSME2, which make up the PA28 regulatory complex (Fig. 5f). There is also a small but significant increase in POMP, which is the maturation factor required for new proteasome assembly. Together, these data suggest that the BRG1-deficient cells have pathway alterations that prime them to better deal with aneuploidy-associated proteotoxic stress.

Immunoproteasome encoding genes are upregulated in response to cellular signals including interferon signalling and oxidative stress[26–28]. We therefore first investigated whether the changes in protein levels at early time points following BRG1 loss (Fig. 5f) were related to changes in gene expression. To investigate this, we analysed mRNA levels of the genes encoding all three immunoproteasome catalytic subunits (*PSMB8*, *PSMB9*, and

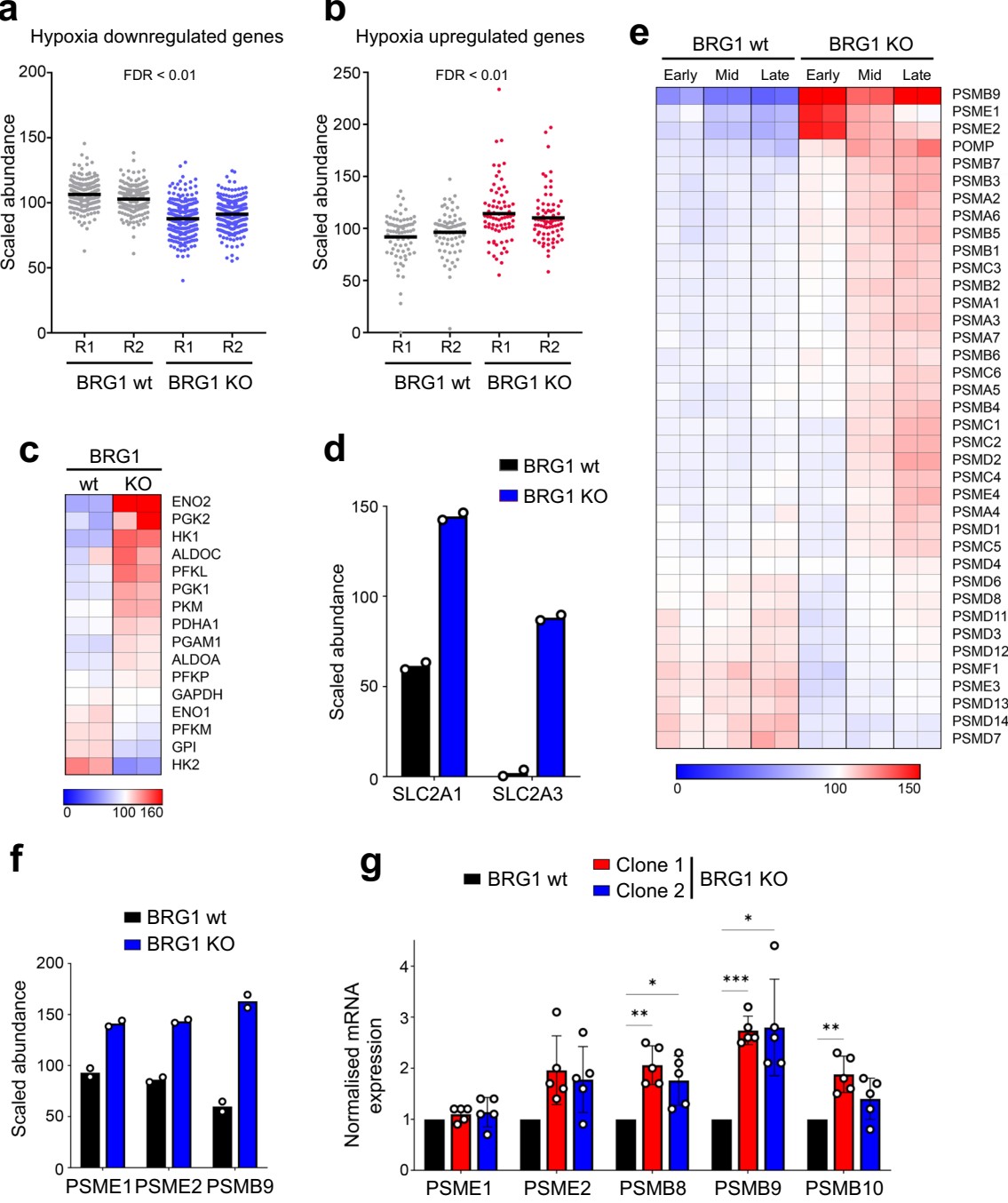

**Fig. 5 BRG1-deficient cells show changes in pathways associated with tolerance to aneuploidy at early time points. a** Scaled abundance of hypoxia proteins with a downregulation trend (GSEA gene list MANALO_HYPOXIA_DOWN) by proteome analysis in HCT116 and BRG1 knockout (KO) cells at early time points. Duplicates are represented as R1 and R2. **b** Scaled abundance of hypoxia proteins with upregulation trend (GSEA gene list MANALO_HYPOXIA_UP) by proteome analysis in HCT116 and BRG1 KO cells at early time points. Duplicates are represented as R1 and R2. **c** Heatmap showing a scaled abundance of glycolysis proteins in HCT116 and BRG1 KO cells at early time points. Duplicates are shown side by side. The colour key indicates the relative abundance of each protein (scale: 0–160). **d** Scaled abundance of SLC2A1 and SLC2A3 glucose importers by proteome analysis in HCT116 and BRG1-deficient cells at early time points. Data were presented as the mean; $n = 2$. **e** Heatmap showing scaled abundance of proteasome proteins in HCT116 and BRG1-deficient (KO) cells at early, mid and late time points. Duplicates are shown side by side. The colour key indicates the relative abundance of each protein (scale: 0–150). **f** Scaled abundance of immunoproteasome subunits by proteome analysis in HCT116 and BRG1-deficient (KO) cells at early time points. Data were presented as the mean; $n = 2$. **g** mRNA expression of immunoproteasome subunits in HCT116 and two BRG1-deficient (KO) clones by RT-PCR analysis. The values have been normalised to GAPDH expression. Data were presented as the mean ± SD; $n = 5$. The $p$ value was calculated with two-way ANOVA-Dunnett. *$p < 0.05$, **$p < 0.01$ and ***$p < 0.001$.

*PSMB10*; note that only PSMB9 was identified in the proteomic analysis) as well as the two PA28 subunits (*PSME1* and *PSME2*) in both BRG1 KO clones by RT-PCR. We found that mRNA levels of these genes were significantly upregulated in at least one of the knockout clones relative to the parental HCT116 cells with the exception of *PSME1* (Fig. 5g).

STAT1 and STAT2 are activators of interferon-regulated transcription, so we looked to see whether upregulation of these factors might be responsible for upregulation of the immunoproteasome and PA28 encoding genes in the BRG1-deficient cells. We found that STAT1 and STAT2 protein levels were substantially upregulated in both BRG1 KO clones (Supplementary Fig. 6a). We also detected elevated *STAT1* and *STAT2* mRNA levels in both BRG1 KO clones (Supplementary Fig. 6b), and following BRG1 depletion using siRNA (Supplementary Fig. 6c), suggesting that STAT1 and STAT2 could be driving increased transcription of downstream immunoproteasome encoding genes.

**Cells lacking BRG1 show increased survival and elevated aneuploidy following treatment with Mps1 inhibitors**. Together, these data are consistent with a model in which BRG1 loss leads to changes in pathways that promote tolerance to aneuploidy. To test whether the BRG1-deficient cells are indeed more tolerant to aneuploidy, we used Mps1 inhibitors to drive chromosome missegregation. Under these conditions, this model predicts that the absence of BRG1 will allow cells that have undergone a nonlethal chromosome missegregation event to survive without a fitness penalty and therefore persist in the population. To test this, we used both BRG1 KO clones at an early passage, prior to any chromosome gains or BRG1 re-expression.

Cells were treated with increasing doses of two different Mps1 inhibitors (Reversine or AZ3146) and viable cells were quantified on day 5. We found that both BRG1 KO clones were able to survive better in the presence of either drug across multiple doses when compared with the parental HCT116 cell line (Fig. 6a). We chose two doses of each drug and performed a time course where viability was monitored. Again, we found that both BRG1 KO clones survived proportionately better than the HCT116 parental cell lines when treated with either Reversine or AZ3146 (Fig. 6b). These data were consistent with the possibility that chromosome missegregation events are better tolerated in the BRG1 KO cells.

While consistent with increased tolerance to aneuploidy, there are other mechanisms by which the BRG1 KO cells could be more resistant to Mps1 inhibitors, such as altered drug uptake. We, therefore, wanted to look at the karyotypes of the Mps1 inhibitor-treated cells to determine whether the BRG1 KO cells showed an elevated frequency of aneuploidy, which would be the prediction if the improved survival in the KO cells is related to increased tolerance. To do this, we modified the experimental set-up to monitor the frequency of aneuploidy in surviving cell population (Fig. 6c). Again, we used two different Mps1 inhibitors, Reversine and AZ3146, and we chose two doses of each. Cells were first treated with either Reversine or AZ3146 for a period of 96 h to induce chromosome missegregation, and then cell populations were expanded in the absence of the inhibitors to ensure that we were measuring ploidy in viable, surviving cells (Fig. 6c). Mitotic spreads were prepared and chromosome numbers counted (Supplementary Fig. 6d). In doing so, we found that the surviving cells from the two BRG1 KO clones were significantly more likely to have chromosome gains when compared with the parental HCT116 cells (Fig. 6d). Moreover, this was apparent after treatment with either Reversine or AZ3146 and at both doses tested (Fig. 6d). Together, these data support the conclusion that BRG1 loss in HCT116 cells leads to greater tolerance of aneuploidy.

**Cancer samples with loss of BRG1 show elevated levels of aneuploidy**. These data raise the possibility that cancer samples with loss of BRG1 might show elevated levels of aneuploidy compared with BRG1-proficient cancers. To test this, we used copy number segment data to calculate mean absolute change in ploidy (as a deviation from 0, which represents normal ploidy) in cancer samples harbouring mutations in the *SMARCA4* gene, which encodes BRG1. As a positive control, we also analysed samples stratified by their TP53 status, loss of which is known to correlate with increased levels of aneuploidy[18,30].

Because these studies were performed in HCT116 cells, which are a human colon cancer cell line, we first looked at aneuploidy levels in colon cancer samples (COAD). We found that many of the BRG1 (SMARCA4) mutant samples are aneuploid, but that the magnitude of the change from normal ploidy is much less than that of the BRG1-proficient samples (Supplementary Fig. 7a). This contrasts with the pattern seen when p53 deficient samples are compared with p53 proficient samples (Supplementary Fig. 7b), and does not support a role for BRG1 loss leading to aneuploidy tolerance in colorectal cancer. We nevertheless considered the possibility that the gain of chr18 in this model system is relevant. In BRG1-proficient colon cancers, there is a frequent loss of chr18. In contrast, the BRG1-deficient colon cancers rarely show chr18 loss (Supplementary Fig. 7c). SWI/SNF mutations tend to occur in colon cancers with microsatellite instability (MSI), but not in colon cancers with CIN where chr18 is frequently lost. Based on the data here, one interpretation of this pattern is that loss of both BRG1 and chr18 is deleterious in colon cells.

We additionally interrogated lung adenocarcinoma (LUAD) samples, where *BRG1* (*SMARCA4*) is more commonly mutated. Interestingly, we find that the absolute ploidy change is greater in the BRG1-deficient samples when compared with the BRG1-proficient samples (Fig. 7a) similar to the difference in ploidy between p53 proficient and deficient samples (Fig. 7b). We also found a statistically significant increase in absolute ploidy change in renal papillary cell carcinoma (KIRP; Fig. 7c). To exclude potential confounding effects, this analysis was done after excluding samples with mutations in TP53, although interestingly, we note that p53 mutations correlate with decreased aneuploidy scores in KIRP (Fig. 7d).

Notably, we found that there is a statistically significant anti-correlation between *SMARCA4* (BRG1) mRNA expression levels and those of the immunoproteasome encoding genes in lung cancer samples (Fig. 7e–I and Supplementary Table 1), suggesting that loss of BRG1 leads to upregulation of immunoproteasome gene expression in cancer. In contrast, we don't see a relationship between BRG1 expression and these genes in colorectal cancer (Supplementary Table 1), where there is no evidence of increased aneuploidy in the absence of BRG1. In lung cancer, therefore, loss of BRG1 is associated with increased immunoproteasome expression and increased levels of aneuploidy, which is consistent with the possibility that changes in tolerance pathways due to BRG1 loss might contribute to the survival of aneuploid cells during cancer progression.

## Discussion

Here, we find that loss of BRG1 in HCT116 cells leads to a decrease in fitness and to alterations in pathways that impact on tolerance of aneuploidy. Over time, we find that the KO cells evolve and show improved fitness, which correlates with a gain in copies of chromosome 18. In addition, when challenged with inhibitors that drive chromosome missegregation, the KO cells survive better, and the survivors show a greater incidence of aneuploidy when compared with the parental cells. These data

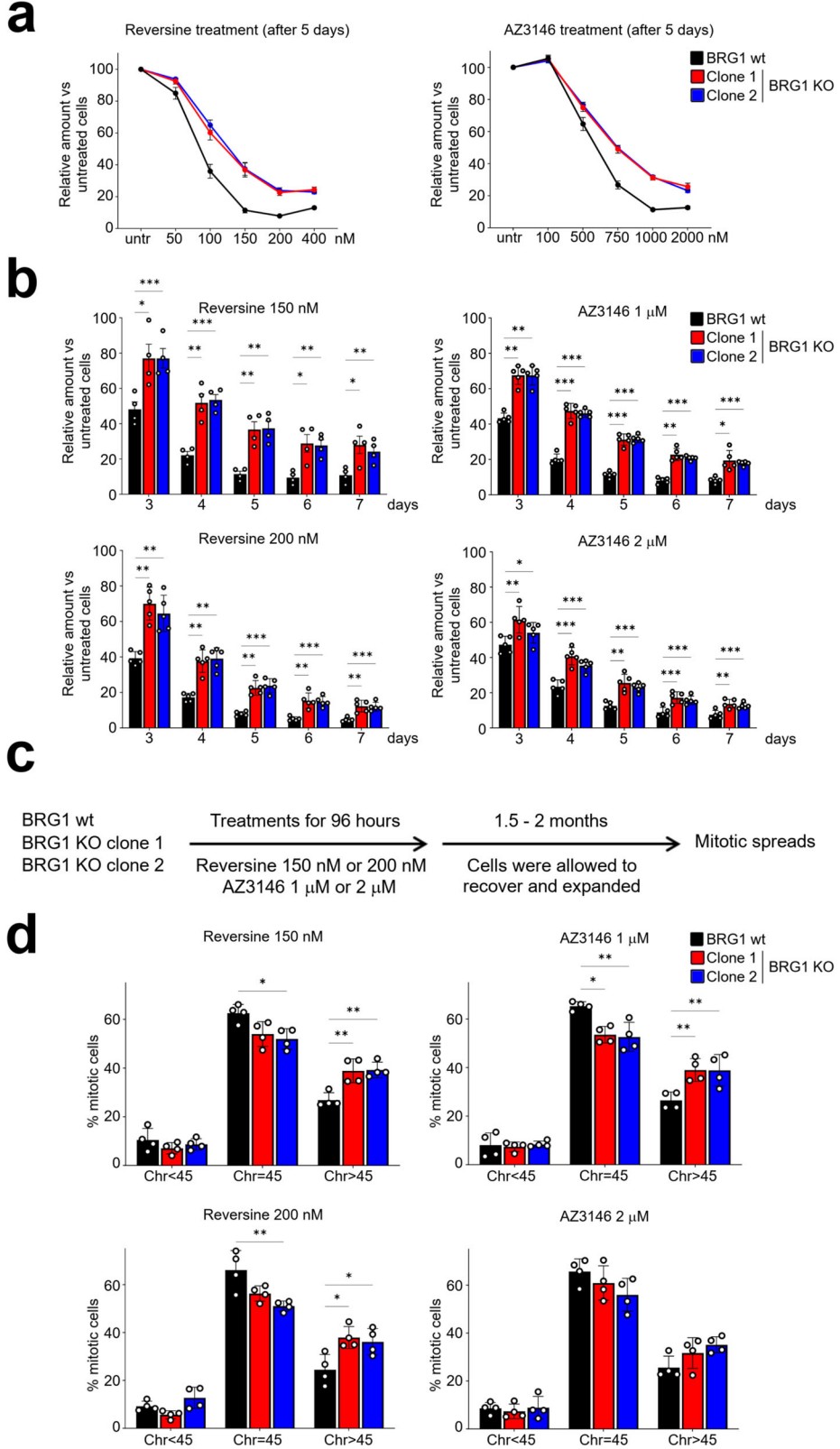

were consistent with a model in which the early changes in aneuploidy-tolerance pathways allow the BRG1 knockout cells to explore different karyotypes without a fitness penalty. When a karyotypic change occurs that gives the cells a fitness advantage, this is selected for in the population. It is possible that this accounts for the aneuploidy present in the *Baf180* knockout mouse ES cell line[10].

Subunits of the SWI/SNF complexes are frequently lost in cancer cells, suggesting that this might be relevant to tumour-igenesis and the ability of SWI/SNF-deficient tumour cells to tolerate aneuploidy. It will be important to establish how universal the impact of BRG1 loss on aneuploidy-tolerance pathways is. In that regard, it is interesting to note BRG1 loss has also been shown to lead to increased glycolysis in glioma-initiating cells

**Fig. 6 BRG1-deficient cells show increased survival and aneuploidy after Mps1 inhibitor-induced chromosome missegregation. a, b** Relative cell viability in HCT116 or BRG knockout (KO) cells. Cells were exposed to the indicated concentrations of Reversine or AZ3146 for 7 days and cell viability was monitored using the sulforhodamine B (SRB) assay and normalised to untreated cell survival. Line graphs in **a** show the relative cell viability after 5 days of exposure to Reversine (left panel) or AZ3146 (right panel). Bar graphs in **b** show survival over time following exposure to 150 or 200 nM Reversine (left panels) or 1 or 2 μM AZ3146 (right panels). Data were presented as the mean ± SEM; $n = 4$. The $p$ value was calculated using two-way ANOVA-Dunnett. $*p < 0.05$, $**p < 0.01$, $***p < 0.001$. **c** Experimental design for monitoring aneuploidy in surviving cell populations after chromosome missegregation induced by Mps1 inhibitor exposure. **d** Metaphase spreads prepared after treatment outlined in **c** were analysed, and the number of chromosomes per metaphase was quantified. The percentage of cells with chromosome gain (Chr >45), loss (Chr <45) or no change (Chr = 45) was calculated. Between 65 and 145 metaphases were analysed per sample per experiment. Data were presented as the mean ± SD; $n = 4$. The $p$ value was calculated with two-way ANOVA-Dunnett assuming sphericity. $*p < 0.05$ and $**p < 0.01$.

albeit through a different mechanism[31]. In addition, we found loss of SWI/SNF subunits BAF180 (PBRM1), BAF200 (ARID2) and BAF250A (ARID1A) had altered levels of glycolysis pathway components[6]. Moreover, the BAF180 subunit of SWI/SNF has been implicated in the regulation of hypoxia- and HIF1-dependent gene expression in renal cells, and the data suggest that loss of BAF180 (together with VHL inactivation) leads to upregulation of glycolysis[32,33]. In the progression of renal cancer, cells are already aneuploid at the point when BAF180 is lost[34], and one interesting possibility is that BAF180 loss can contribute to tumourigenesis by improving the competitive fitness of these aneuploid cells.

In contrast, however, loss of BRG1 in lung cancer cells led to reduced glycolysis and increased dependence on oxidative phosphorylation[35], which would not be consistent with increased aneuploidy tolerance. It is possible that the changes observed in the model cell system used here are not widely applicable to BRG1 loss in different tissue types and genetic backgrounds. Alternatively, the increased aneuploidy that is apparent in BRG1-deficient lung cancer samples may relate more to changes in the proteasome. The most dramatic changes observed in the proteasome composition upon loss of BRG1 are in components of the immunoproteasome. This complex is upregulated in response to interferon signalling or oxidative stress and allows proteins to be targeted for ubiquitin-independent degradation[26–28]. Upregulation of the immunoproteasome will allow cells to better cope with increased levels of unfolded or misfolded proteins, and consequently will prime the cell to tolerate copy number alterations and aneuploidy.

Upregulation of the immunoproteasome subunits will also impact the immunopeptidome of the cells, and interestingly, the SWI/SNF subunits BAF200 (ARID2), BAF180 (PBRM1) and BRD7 were identified in a screen for resistance to T cell-mediated killing[36]. Moreover, the SWI/SNF-deficient cells showed enhanced responsiveness to interferon signalling, which upregulates the immunoproteasome, suggesting this could be a common feature of SWI/SNF-deficient cells. Recently, overexpression of the immunoproteasome subunits PSMB8 and PSMB9 was shown to increase immunogenicity and lead to a better response to immune checkpoint inhibitor therapy[37]. Notably, patients with inactivating mutations in SWI/SNF genes in ccRCC show better response to immune checkpoint inhibitor therapy in at least some studies[38]. We are currently investigating whether alterations in immunoproteasome levels in SWI/SNF-deficient cancers contribute to checkpoint inhibitor response.

In the HCT116 model system, we found that chromosome gains correlate with increased fitness in the BRG1 knockout cells. Aneuploidy is normally associated with a loss of fitness[7], including in the HCT116 cell line[2]. However, aneuploidy has been shown to be advantageous under particular conditions. For example, colorectal cancer cells that have an additional chromosome grow better under conditions of limiting nutrients, 5-FU, and hypoxia than their diploid counterparts[39]. In addition,

aneuploidy in liver cells appears to allow adaptation to chronic injury[40]. These results are likely due to two factors. First, upregulation of pathways related to the tolerance of proteotoxic and metabolic stress will provide an advantage to aneuploid cells when growth conditions become stressful. Second, in the case of mixed populations of aneuploid cells, aneuploidy provides a genetic variation that allows rapid adaptive evolution in response to new environments.

The data here suggest that in BRG1 knockout cells, aneuploidy can not only be tolerated but can lead to a relative gain in fitness. Our data suggest that by upregulating pathways that mitigate against proteotoxic and metabolic stress, the cells are primed to tolerate changes in the karyotype. Subsequently, because the loss of BRG1 leads to a proliferative defect in this cell line, the ability to explore new karyotypes without a fitness penalty allows adaptive evolution. This provides an insight into the potential relationship between SWI/SNF activity and aneuploidy in cancer cells, and this gives us a useful tool to interrogate pathways of evolution.

## Methods

**Generation of BRG1-deficient HCT116 cell lines.** BRG1 knockout cell lines were obtained at the same time that tunable SMASh-tagged BRG1 cells were generated. HCT116 cells were co-transfected in a ratio 3:1 with a recombination template (pBS BRG1 SMASh tag), and a plasmid encoding the endonuclease Cas9 and a gRNA targeting BRG1 gene (5′-caccgGGCCGAGGAGTTCCGCCCAG-3′, 5′-aaacCTGGGCGGAACTCCTCGGCCc-3′) just after the initial codon (pSpCas9n(BB)-2A-puro). Cells were transfected using Lipofectamine LTX with Plus reagent (Invitrogen) following the manufacturer's recommendation and clones were selected in the presence of 800 mg/mL Geneticin (G418 Sulfate, Gibco) in the growing medium. Clonal cell populations were screened by Western blotting and BRG1 deletion was confirmed by sequencing and in clone 2 by proteomic analysis.

**siRNA mediated depletion.** Around 800,000 cells per condition were transfected at the same time that they were seeded (6-cm dish) using Lipofectamine RNAi-MAX Transfection Reagent following manufacturer instructions. Non-targeting control or siRNAs (single or combinations) targeting BRG1 were used (Supplementary Table 2).

**Clonogenic assays.** Analysis of colony formation was measured by clonogenic assay. For each condition, 300 cells were plated in 6 cm plates. To assess changes in colony formation efficiency during cellular evolution, 1000 cells were seeded in six-well plates in triplicate. Cells were kept in culture for 10 days and colonies were fixed and stained with 0.4% methylene blue in 50% methanol.

**Proliferation and viability assays.** One million cells were seeded in 10 cm plates and cells were trypsinized and counted every 2 days. Once counted, cells were reseeded in new dishes with appropriate dilutions. Alternatively, cell viability was assessed with CellTiter-Glo® Luminescent Cell Viability Assay (Promega). Cells were seeded in triplicate at 1500 cells per well in 96 well plates. Chemoluminescence was monitored at 24, 48, 72 and 96 h from seeding.

Growth in the presence of Mps1 inhibitors Reversine (R3904, Sigma-Aldrich) and AZ3146 (SML1427, Sigma-Aldrich) were tested using the SRB (Sulforhodamine B, 230162, Sigma-Aldrich) assay. About 1500 or 4500 cells per well for parental HCT116 or BRG1 KO clones, respectively, were seeded in 96 well plates, using five wells for each condition. After 24 h, the Mps1 inhibitors were added to the indicated concentration. Cell proliferation was measured at 3, 4, 5, 6 and 7 days after the addition of the inhibitor. Cells were fixed by the addition of a volume of cold 10%

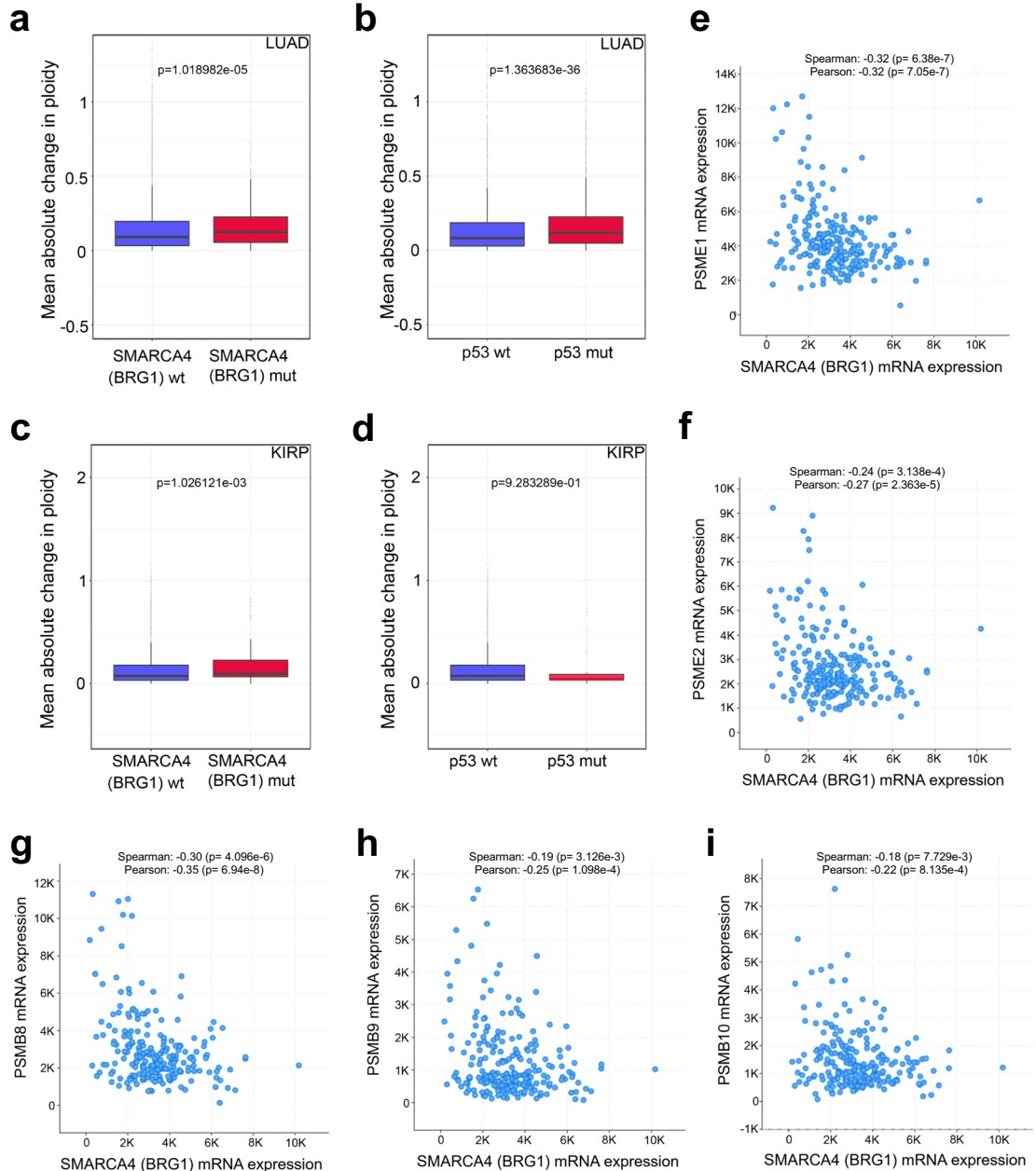

**Fig. 7 Cancer samples with mutations in BRG1 (SMARCA4) have higher aneuploidy scores than samples with wild-type BRG1 and expression of immunoproteasome genes is inversely correlated with SMARCA4 expression. a**, **b** Mean absolute change in ploidy was calculated for lung adenocarcinoma (LUAD) samples ($n = 20,720$) analysed according to BRG1 (SMARCA4) mutational status (**a**). Mean ploidy change is 0.138 (wt BRG1) and 0.159 (mutant BRG1). TP53 mutant samples were excluded from the analysis in **a** to avoid confounding effects. TP53 mutant samples were analysed separately as a point of comparison (**b**). Mean ploidy change is 0.131 (wt p53) and 0.161 (mutant p53). The mean absolute change in aneuploidy was calculated as a deviation from diploid (zero), and *p* values (indicated) were calculated with a two-sided Welch's *t*-test. **c**, **d** Mean absolute change in ploidy was calculated for kidney renal papillary cell carcinoma (KIRP) samples ($n = 11,600$) analysed according to BRG1 (SMARCA4) mutational status. (**c**). Mean ploidy change is 0.136 (wt Brg1) and 0.171 (mutant Brg1). TP53 mutant samples were excluded from the analysis in **d** to avoid confounding effects. TP53 mutant samples were analysed separately as a point of comparison (**d**). Mean ploidy change is 0.137 (wt p53) and 0.139 (mutant p53). The mean absolute change in aneuploidy was calculated as a deviation from diploid (zero), and *p* values (indicated) were calculated with a two-sided Welch's *t*-test. **e**–**i** BRG1 mRNA levels show a significant negative correlation with mRNA expression of PA28 (**e**, **f**) and immunoproteasome (**g**, **h**, **i**) genes in cancer samples. Analysis was performed on cBioportal using the TCGA lung adenocarcinoma (LUAD) dataset, and *p* values derived from Spearman or Pearson correlation analysis are indicated on each panel.

TCA (Trichloroacetic acid solution, T0699, Sigma-Aldrich) and incubated for at least 30 min at room temperature. Plates were washed five times using distilled water and allowed to dry at room temperature. Cells were incubated with SRB solution (0.057% in 1% acetic acid) for an hour at room temperature followed by five quick washes with 1% acetic acid. Excess liquid was removed and plates were allowed to dry at room temperature. SRB was solubilized by the addition of Tris 10 mM (pH 10.1) and incubated for 10 min at room temperature in a shaker.

Absorbance at 490 nm was measured using a plate reader (SpectraMax iD5, Molecular Devices). Proliferation was calculated relative to the corresponding untreated HCT116 parental cell line or BRG1 KO clone for each day.

**Flow cytometry.** Cell cycle profiles were analysed by flow cytometry. Cells were trypsinized, washed twice with PBS and fixed with 70% ethanol overnight at

−20 °C. The samples were stained with 5 µg/ml propidium iodide and 0.1 mg/ml RNaseA in PBS for 30 min. Analytic flow profiles of DNA content were recorded in a BD LSR II Flow Cytometer (BD Biosciences) machine using BD FACSDiva v9.0. A minimum of 10,000 events per sample was counted and data were analysed with Flowjo v10.1 software.

**Protein extracts and Western blotting.** Cell pellets were lysed in 50 mM Tris pH 7.9/8 M Urea/1%Chaps and incubated at 4 °C with agitation for at least 30 min. The lysate was cleared by centrifugation and the supernatant was collected. The protein concentration of the extracts was measured by Bradford assay. About 25 µg of supernatants were resolved by SDS-polyacrylamide gel electrophoresis and transferred onto a Hybond-C Extra Nitrocellulose membrane (Fisher Scientific UK, Loughborough, UK). The membrane was blocked in 5% milk, 0.1% Tween-20 in TBS buffer for 1 h and probed overnight with antibodies against BRG1 (Santa Cruz, sc-17796, dilution 1:2000), a-tubulin (Abcam Ab7291, dilution 1:5000), p21 (Cell Signalling, 2947 S, dilution 1:2000), INO80 (Bethyl, A303-371A, dilution 1:2000) in 5% milk, 0.1% Tween-20 in TBS buffer. The membrane was washed three times with 0.1% Tween-20 in TBS and incubated for 1 h with horseradish peroxidase (HRP)-conjugated secondary antibodies in 5% milk, 0.1% Tween-20 in TBS. The secondary antibodies used were Rabbit anti-Mouse HRP (Dako, P0260, dilution 1:5000) and Goat anti-Rabbit HRP (Dako, P0448, dilution 1:5000). Proteins were visualised by using in-house ECL reagent or SuperSignal West Pico Chemiluminescent Substrate (Life Technologies).

**Metaphase fluorescent in situ hybridisation (FISH).** Cells were treated with 0.1 µg/ml colcemid (Gibco) for 3 to 4 h. Subsequently, cells were trypsinized, washed twice with PBS and resuspended in hypotonic buffer (0.075 M potassium chloride). After 15 min incubation at 37 °C, cells were centrifuged and the pellet was gently resuspended in 3:1 methanol/glacial acetic acid fixative solution and incubated for 20 min at room temperature. Cells were then washed twice with fixative and a few drops were released onto an alcohol-cleaned slide and allowed to air dry. FISH was performed according to the MetaSystems Probes recommended protocol. Slides were incubated for 20 min in 2X SSC at 37 °C and dehydrated in ethanol series (70, 85 and 100%), 2 min in each solution. Probes XA 21q22 and XCE 18 Green (MetaSystems) were used to detect copy number variation for chromosome 21 and chromosome 18. Probes were applied onto the slide and the samples were denatured on a hotplate at 75 °C for 2 min. Slides were then incubated in a humid chamber overnight at 37 °C. Next, slides were washed in 0.4X SSC pH 7.0 at 72 °C for 2 min and later in 2X SSC, 0.05% Tween-20 pH 7.0 for 30 s at room temperature. Once dry, slides were stained with DAPI for 10 min. Metaphase spreads and FISH signals were captured on a confocal microscope using SlideBook 6 and quantified using Fiji/ImageJ 1.53c.

**Metaphase chromosome counting.** Cells were treated with 0.1 µg/ml colcemid (Gibco) for 4 h. Subsequently, cells were trypsinized, washed twice with PBS and resuspended in hypotonic buffer (0.075 M potassium chloride). After 15 min incubation at 37 °C, cells were centrifuged and the pellet was gently resuspended in 3:1 methanol/glacial acetic acid fixative solution and incubated for 20 min at room temperature. Cells were then washed twice with fixative and a few drops were released onto an alcohol-cleaned slide and allowed to air dry. Chromosomes were stained with DAPI-containing mounting media (P36931, ProLong Gold Antifade Mountant with DAPI, Invitrogen).

Where cells were treated with Mps1 inhibitors, HCT116 parental cells or BRG1 clones 1 and 2 were seeded in 10 cm plates and incubated with two different concentrations of Reversine (150 or 200 nM) or AZ3146 (1 or 2 µM) for 96 h. Plates were washed three times and cells were allowed to recover and expanded for a period of 1.5 to 2 months after the Mps1 inhibitor exposure. Images were captured using an EVOS XL Core Imaging System (Invitrogen). Between 65 and 145 metaphases were analysed per condition, and four independent experiments were carried out.

**Analysis of mitotic events.** For micronuclei analysis, cells were seeded on a glass coverslip and were kept in culture for 3 days. Cells were then fixed with 4% paraformaldehyde in PBS and permeabilised with 0.5% Triton-X100 in PBS. Nuclei were counterstained with DAPI for 10 min and mounted with Vectashield mounting media. Micronuclei of at least 800 cells were manually counted.

For the analysis of mitotic defects, cells were synchronised with 6 µM RO-3306 for 6 h and released by washing three times with PBS. After 20 min from the release, mitotic cells were collected by vigorously shaking the plate. These cells were then reseeded on poly-lysine treated coverslips and fixed with 4% paraformaldehyde in PBS at different time points (20, 30 and 40 min following mitotic shake-off). Cells were permeabilised with 0.5% Triton-X100 in PBS, stained with anti-PICH antibody (Millipore 04-1540, dilution 1:1000) overnight and the nuclei were counterstained with DAPI.

**Array-based comparative genomic hybridisation (CGH) analysis.** To study copy number variation in HCT116, cells were trypsinized and washed twice with PBS. The CGH array service (KaryoNIM Stem Cells) by NIMGenetics was used for CGH array analysis.

**RT-qPCR analysis of immunoproteasome component expressions.** Between $2 \times 10^6$ and $5 \times 10^6$ cells were collected per sample and RNA was extracted using an RNAeasy Mini Kit (74106, QIAGEN), according to the manufacturer's protocol. cDNAs were synthesised from 1 µg RNA using Superscript II First Strand RT-PCR system (11904018, Invitrogen), following the manufacturer's protocol. cDNAs were diluted 1:10 and quantitative PCR was performed using a StepOnePlus Real-Time PCR System (Applied Biosystems) on 10 µL volume reactions prepared with Power SYBR green PCR master mix (4367659, Applied Biosystems). A minimum of three independent experiments were carried out, and each sample was run in triplicate per experiment. GAPDH expression was used for normalisation. Primer sequences used for PSMB8, PSMB9, PSMB10, PSME1, PSME2, STAT1, STAT2 and GAPDH expression analysis are listed in Supplementary Table 2.

**Statistical analyses.** Analyses of quantitative cell biology data (proliferation, survival, cell cycle profiles, micronuclei analysis, metaphase spreads, RT-qPCR and FISH) were performed using GraphPad Prism 9.3.0. Raw data, statistical tests used and $p$ values are provided in the Source Data file.

**Proteomic analyses**

*Sample preparation and TMT labelling.* Cell pellets were dissolved in 150 µL lysis buffer containing 1% sodium deoxycholate (SDC), 100 mM triethylammonium bicarbonate (TEAB), 10% isopropanol, 50 mM NaCl and Halt protease and phosphatase inhibitor cocktail (100X) (Thermo, #78442) with pulsed probe sonication for 15 s. Samples were boiled at 90 °C for 5 min and were re-sonicated for 5 s. Protein concentration was measured with the Coomassie Plus Bradford Protein Assay (Pierce) according to the manufacturer's instructions. Aliquots of 100 µg of total protein were taken for trypsin digestion and the concentrations were equalised. Samples were reduced with 5 mM tris-2-carboxyethyl phosphine (TCEP) for 1 h at 60 °C and alkylated with 10 mM Iodoacetamide (IAA) for 30 min in dark. Proteins were then digested overnight by trypsin (Pierce) at 75 ng/µL. The peptides were labelled with the TMT10plex multiplex reagents (Thermo) according to manufacturer's instructions and were combined in equal amounts to a single tube. The combined sample was then dried with a centrifugal vacuum concentrator.

*Basic reverse-phase peptide fractionation and LC-MS analysis.* Offline high pH Reversed-Phase (RP) peptide fractionation was performed with the XBridge C18 column (2.1 × 150 mm, 3.5 µm, Waters) on a Dionex Ultimate 3000 HPLC system. Mobile phase A was 0.1% ammonium hydroxide and mobile phase B was 100% acetonitrile, 0.1% ammonium hydroxide. The TMT labelled peptide mixture was fractionated using a multi-step gradient elution method at 0.2 mL/min as follows: for 5 min isocratic at 5% B, for 35 min gradient to 35% B, gradient to 80% B in 5 min, isocratic for 5 min and re-equilibration to 5% B. Fractions were collected every 42 s and vacuum dried.

LC-MS analysis was performed on the Dionex Ultimate 3000 system coupled with the Orbitrap Fusion Lumos Mass Spectrometer (Thermo Scientific). Each peptide fraction was reconstituted in 40 µL 0.1% formic acid and 7 µL were loaded to the Acclaim PepMap 100, 100 µm × 2 cm C18, 5 µm, 100 Å trapping column at 10 µL/min flow rate. The sample was then subjected to a gradient elution on the EASY-Spray C18 capillary column (75 µm × 50 cm, 2 µm) at 45 °C. Mobile phase A was 0.1% formic acid and mobile phase B was 80% acetonitrile, 0.1% formic acid. The gradient separation method at a flow rate 300 nL/min was as follows: for 90 min gradient from 5 to 38% B, for 10 min up to 95% B, for 5 min isocratic at 95% B, re-equilibration to 5% B in 5 min, for 10 min isocratic at 5% B. The precursor ions between 375–1500 m/z were selected with a mass resolution of 120k, AGC $4 \times 10^5$ and IT 50 ms for CID fragmentation with isolation width 0.7 Th in the top speed mode (3 s). The collision energy was set at 35% with AGC $1 \times 10^4$ and IT 50 ms. MS3 quantification was obtained with HCD fragmentation of the top five most abundant CID fragments isolated with Synchronous Precursor Selection (SPS). Quadrupole isolation width was set at 0.7 Th, collision energy was applied at 65% and the AGC setting was $1 \times 10^5$ with 105 ms IT. The HCD MS3 spectra were acquired for the mass range 100–500 with 50k resolution. Targeted precursors were dynamically excluded for further isolation and activation for 45 s.

*Database search and protein quantification.* The SequestHT search engine in Proteome Discoverer 2.2 (Thermo Scientific) was used to search the raw mass spectra against reviewed UniProt human protein entries for protein identification and quantification. The precursor mass tolerance was set at 20 ppm and the fragment ion mass tolerance was 0.02 Da. Spectra were searched for fully tryptic peptides with maximum of two missed-cleavages. TMT6plex at N-terminus/K and Carbamidomethyl at C were defined as static modifications. Dynamic modifications were oxidation of M and Deamidation of N/Q. Peptide confidence was estimated with the Percolator node. Peptide FDR was set at 0.01 and validation was based on q-value and decoy database search. The reporter ion quantifier node included a TMT10plex quantification method with an integration window tolerance of 15 ppm and an integration method based on the most confident centroid peak at the MS3 level. Only unique peptides were used for quantification, considering protein groups for peptide uniqueness. Peptides with average reporter signal-to-noise >3 were used for protein quantification.

*Proteomic bioinformatics analysis.* Differential expression and pathway analysis of proteomics data were performed in Perseus 1.6 software[41] using the ANOVA test and the 1D annotation enrichment method[42], respectively. The ANOVA test was done using scaled protein abundances across samples whereas log2ratios were used for pathway enrichment analysis using biological terms from GSEA, KEGG, GOBP-slim and CORUM resources. All results were filtered for Benjamini–Hochberg FDR <0.05. Visualisation of pathway analysis was done in RStudio v3.6.1 with the ggplot2 library. Principal Component Analysis was done in RStudio v3.6.1. Heatmaps were visualised in the online clustering tool Phantasus v1.11.0 (https://artyomovlab.wustl.edu/phantasus/).

*Cancer database bioinformatics analysis.* Copy number segment data was sourced via the Genomics Data Commons portal[43] for kidney, breast, lung and colon tissues. Data were filtered to remove all segments with a length of under ~14,000 bp (the bottom quartile across all segment lengths) to better reflect partial ploidy change.

The copy number segment data were processed to provide a fingerprint of mean partial ploidy per chromosome arm for each patient. Mutational data were sourced via COSMIC[44] and patient IDs were matched to ploidy fingerprint patients.

To measure how individual mutations effect arm-wise ploidy change we segmented patients first by sample tissue type and then into mutated and non-mutated class groups. Within these groups mean arm-wise ploidy changes were calculated and compared[45]. To estimate the genome-wide effect of a mutation on ploidy we calculated the mean absolute ploidy for each patient for each group.

Significance values were measured by comparing mean absolute ploidy change for each patient in each group (mutated/non-mutated within each tissue group) via a student *t*-test where more than 30 samples of each group were available.

**Reporting summary**. Further information on research design is available in the Nature Research Reporting Summary linked to this article.

## Data availability

The data that support this study are available from the corresponding author upon reasonable request. The mass spectrometry proteomics data generated in this study have been deposited to the ProteomeXchange Consortium (http://proteomecentral.proteomexchange.org) via the PRIDE partner repository under accession code PXD014954.

The remaining data generated in this study are provided in the Supplementary Information and Source Data. Source data are provided with this paper.

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

## Acknowledgements

We thank all members of the Downs lab for helpful discussions, Nadia Hegerat and Helfrid Hochegger for advice with CRISPR-Cas9 cell line generation and Cornelia Meisenberg for experimental assistance. This study was supported by Cancer Research UK (P.Z-V. and J.A.D.; C7905/A25715), Cancer Research UK Centre grant (T.I.R. and J.S.C; C309/A25144) and the Medical Research Council (G.B-H. and F.M.G.P.; MR/N50189X/1).

## Author contributions

F.S., P.Z-V. and J.A.D. conceived of the study, designed experiments and analysed data. F.S. and P.Z-V. generated reagents and performed experiments. T.I.R., M.P. and J.S.C. designed and performed mass spectrometry experiments and analysed data. G.B-H. and F.M.G.P. performed copy number analysis of cancer samples. J.A.D. wrote the manuscript with assistance from F.S. and P.Z-V. and input from all authors.

## Competing interests

The authors declare no competing interests.
