## [Peer Review File · Nature Communications]

REVIEWER COMMENTS

Reviewer #1 (Remarks to the Author):

Summary. The authors use Crispr to mutate BRG1 in HCT116 cells and create 3 clones. This suppresses proliferation and over time the clones adapt. Two clones appear to restore Brg1 while a third appears to remain BRG1-deficient, but note that the characterisation here is not sufficiently in depth. The clone that remains BRG1-deficient gains copies of chromosome 18, leading the authors to conclude that BRG1 deficiency enhances aneuploidy tolerance, as explicitly stated in the title. Proteomics evokes hypotheses based on deregulation of hypoxia/glycolysis and immunoproteasome pathways. Testing one of these would make the story considerably more compelling for Nat Comms. More problematic is the lack of evidence to support aneuploidy tolerance. Inhibition of BRG1 imposes a fitness cost exerting a strong selective pressure, evidenced by the two reversion events. Therefore, an alternative explanation is that a random missegregation event yields a subclone with extra copies of 18 which can now restore proliferation in the absence of BRG1 and thus outcompete the original euploid cells. To evoke aneuploidy tolerance, the authors would need to experimentally induce chromosome missegregation to generate random aneuploidies and show that aneuploid cells persist in the BRG1-deficient population for longer.

Critique. Aneuploidy is a hallmark of human cancers despite considerable evidence that its effects are mostly detrimental to cellular and organismal fitness (classically known as the aneuploidy paradox). Therefore, understanding the mechanisms that contribute to aneuploidy's origin, such as chromosome instability (CIN), and its tolerance is of critical interest to the cancer biology community. In their manuscript, Schiavoni & Zuazua-Villar et al. propose that the ablation of BRG1 in the microsatellite unstable, chromosomally stable colon cancer cell line HCT116 leads to aneuploidy tolerance over time. Using CRISPR, the authors generate cell lines deficient for BRG1. Two of the three cell lines regain expression of BRG1 over time thus the authors focus on one clone which remains BRG1 deficient. As these cells are passaged, a gain of chr. 18 is noted. Proteomic analysis reveals an upregulation of glycolysis related proteins and an upregulation of the immunoproteasome (IP) at the early timepoint in the BRG1-/- vs WT cells. The former is consistent with previous observations in trisomic cells (Williams et al. (2008), PMID: 18974345, cited). The latter finding is novel and intriguing since the IP has recently been shown to protect thymic epithelial cells from proteotoxic stress a well-established consequence of aneuploidy (PMID: 29186691, omitted). The authors then proceed to interrogate cancer genomics data stratifying for BRG1 mutation status to evaluate if its mutation correlates with aneuploidy in human cancers which holds true in part. While BRG1 deficiency has been known to cause mitotic abnormalities (Bourgo et al. (2009), PMID: 19458193, cited), replication stress (Cohen et al. (2010), PMID: 20571081, omitted) and activation of the decatenation checkpoint (Dykhuizen et al. (2013), PMID: 23698369, omitted), a role for aneuploidy tolerance has not previously been recognized. Therefore, this manuscript presents an interesting dataset that will be appreciated by the field.

In terms of strengths, the manuscript is very well written and easy to follow for the most part. Likewise, the data is presented mostly appropriately, and the figures are well prepared. The choice of model system is clear. The characterization of the mitotic phenotype (or rather absence thereof) in the BRG1mut cells deserves to be commended due to the absence of an easily detected difference. The

observation that the IP is upregulated in BRG1mut, aneuploid cells is highly interesting in terms of aneuploidy tolerance and indeed novel. The proteomic analysis also corroborates previous mRNA expression analyses that suggested a role for hypoxia and glycolysis in aneuploidy tolerance. In terms of weaknesses, the characterization of the BRG1mut cells is insufficient, especially in the absence of the expected mitotic aberrations observed in murine BRG1-deficient models. Aneuploidy tolerance is a strong term that is overstated in this context as the evidence presented only justifies saying that BRG1 deficiency creates a selective pressure that is buffered by a gain of chr. 18. If this is the new angle, it could be tested by mutating BRG1 in already trisomic HCT116 cells (Vasudevan et al, (2020), PMID: 32097652) but this is probably beyond the scope of this manuscript.

Major issues before publication in Nat Comms can be recommended:

1. The clones need to be better characterised with respect to the nature of the BRG1 mutations and reversions.
2. Evidence of bona fide aneuploidy tolerance needs to be provided, or the emphasis of the story is changed to one that focuses on gain of chromosome 18 being able to buffer loss of BRG1 function.
3. The authors need to test whether upregulation of hypoxia/glycolysis and/or immunoproteasome pathways is responsible for fitness restoration in the absence of BRG1.
4. Weave in additional relevant citations highlighted above.

Other issues to address/consider:

Line 72, the mouse ES cell data needs to be rationalised with the new data in the Discussion.

Line 107, explicitly state that by depletion you are referring to an RNAi experiment.

Line 107, call out of supplementary figures 1c-e precedes supplementary figures 1a-b.

Line 113, FACS profiles too harshly gated to support statement.

Line 115, 1e and f somewhat redundant.

Line 124, need to demonstrate that clone 2 does indeed remain “negative”. See major issue above, what is the nature of the initial BRG1 mutation in the three lines? In 1 and 3, it is a simple single nucleotide deletion that reverts back to WT? Does clone 2 have a more complex indel that is harder to revert, explaining why it remains fixed? Or does clone 2 also restore partial BRG1 function via a different mechanism? E.g. a splice variant not detected by the antibody used. Can the mass spec data be mined/interpreted to rule this out? Note the recent issues with BUB1. It is also important to rule out the trivial explanation of outgrowth of wildtype cells due to the original lines not being clonal. I suspect this is not the case otherwise they would have outgrown prior to the initial characterisations but this needs to be formally addressed in the text.

Line 128, 2c would benefit from the comparisons being on the same 6 well plate and quantification.

Line 132, 2e not compelling, are the n=3 biological replicates?

Line 142, panel 3b can you show the actual chromosome numbers rather than gain or loss, i.e. in any given line is it a gain or loss of 1, 2, 3 or more chromosomes? 3f is better. Can you also correlate chr gains and chr 18 gain in each metaphase? It seems FISH was performed on M-phase spreads already.

Line 152, the array CGH data is nice but as presented doesn't really do the data justice. The authors should stick to array CGH as conventional CGH is something different.

Line 166, enhance FISH IF to better display the chromosomes.

Line 181, the PCA shows very nicely the evolution of the cell line.

Line 189, mass spec nicely corroborated westerns showing loss of BRG1 at all timepoints.

Line 208, panels 4c and 4d could be annotated with titles to better display what is being compared.
Line 211, 4d, 5a and 5c are missing statistical analysis of scaled protein abundance.
Line 217, the language is getting a bit sloppy: "When we look more closely at regulators of the G1/S transition, we find that ..."
Line 241, good effort to rule out mitotic errors.
Line 248, need to formally test whether it is aneuploidy tolerance, i.e. put BRG1 mutants through an Mps1 inhibitor washout protocol to induce random aneuploidies and show that they are not as rapidly eliminated compared to wildtype. See major issue above.
Line 253, original work should be cited at least alongside review Thompson & Compton (2010), PMID: 20123995.
Line 262, reference 18 refers to Cyclin D overexpression in tolerance of whole genome duplication in diploid cells, response to tetraploidy might differ from a response to aneuploidy.
Line 267, why not first point out that hypoxia emerges from the mass spec, then note Hif1 signalling was also recently implicated in aneuploidy tolerance?
Line 278, but none of these ideas are tested, would this not be required for a Nat Comms paper? See major issue above.
Line 308, not obvious how mRNA levels were analysed, would be helpful to clarify in the text.
Line 311, PSM genes are upregulated in clones 1 and 2, this is confusing as clone 1 restores BRG1 function. Was this done at an early time point before reversion?
Line 330, as above the authors present a hypothesis but do not test it. Again, see major issue above.
Line 349 should read "did not".
Line 478 should read "manufacturer".
Line 547, bioinformatics analysis of mass spec experiment is missing.

Reviewer #2 (Remarks to the Author):

The authors had previously found that loss of the BAF180 subunit that is specific to the PBAF complex in which BRG1 is the catalytic subunit in U2OS cells leads to chromosome instability and enhanced aneuploidy. Importantly, they found the ES cells lacking Baf180 were aneuploid with two recurring chromosome gains, but had no observable compromise in fitness. These findings suggest that the PBAF complex regulates aneuploidy, and that aneuploid cells can 'adjust' genomically to attain an adequate level of fitness.

Synopsis : Schiavoni et.al's main focus in this paper was exploring what proteins and pathways are dysregulated when BRG1 is knocked down in the context of HCT116 cell line (colorectal cancer cell line); Schiavoni et.al ultimately concluded that the loss of BRG1 and subsequent dysregulation of processes associated with BRG1 expression, allows for tolerance of aneuploidy over 8 months in the HCT116 cell line. To study BRG1's association to aneuploidy tolerance, Schiavoni et.al used CRISPR to inactivate SMARC4 (encodes BRG1) in HCT116 and created 3 different clonal cell lines with 3 individually sgRNAs in

hope of inactivating SMARC4 and therefore knocking down BRG1. They then monitored these cell lines in-vitro for 8 months, and analyzed the effects of BRG1 at three time points: early, middle (4 months), and late (8 months).

Schiavoni et.al successfully showed that when BRG1 is knocked down: (1) it causes a decrease in rate of proliferation compared to controls which is supported by, (2) a cell count assay, and flow cytometry assay showing that BRG1 KD cells move more slowly in the G1 phase compared to control cells. To ultimately test the effect of BRG1 KD on aneuploidy, they (3) used metaphase FISH with a probe against chromosome 18, and showed that BRG1 KD cells in middle and late phases had a gain in chromosome 18 compared to controls (Figure 3d). Interestingly, they saw that late phase cells with BRG1 KD slowly regained fitness, which was assessed through proliferation rate (cell count) and cell cycle analysis using flow cytometry, which showed that late phase BRG1 KD cells had similar cell cycle profile to that of the control cells.

To determine whether BRG1 causes aneuploidy vs sets the stage for aneuploidy, they assessed chromosomal instability via quantifying micronuclei during cell division, and found no significant difference in the number of micronuclei during division between KD and WT BRG1 in early, middle, or late phase. From these results, they concluded that loss of BRG1 does not directly cause aneuploidy, but rather, may set the stage for aneuploidy tolerance. They coupled these results to proteome and RNA_seq analysis where they first (1), compared WT to early phase KD cells, and then analyzed these same pathways and proteins that were deregulated in early phase and (2) compared early phase and late phase BRG1 KD cells. From their results, Schiavoni et.al postulate that some of these processes (i.e. increase in glycolysis and hypoxia associated proteins and pathways) may confer aneuploidy tolerance. The culmination of this work gave life to their title that loss of BRG1 and its subsequent dysregulated processes may confer aneuploidy tolerance which could potentially set the stage for oncogenesis.

Concerns:

1. The initial work using Baf180 deficient ES cells was based on strong data since it used normal cells; here the work is much less convincing since it is based on a single human colorectal cancer cell line.
2. BRG1 is present in multiple complexes, so it is not at all clear that is working through PBAF, as appears to be assumed here.
3. The authors discuss apoptosis, and do not observe it upon BRG1 depletion. Was cellular senescence considered?
4. While it was great that three subclones were initially established, two of these clones regained BRG1 expression. Therefore, the work is based on only a single subclone.
5. It is not clear whether gain of chromosome 18 was one of the two chromosome gains found in BAF180 deficient U2OS cells.
6. When discussing the mouse ES cells lacking Baf180, it would be preferable to use "ES" cells rather than "MES."

7. Please use Baf180 instead of BAF180 to designate mouse.

8. Referring to the “BRG1 null cell lines” (for example line 111) is incorrect, as “null” refers to alleles, but here you are discussing the protein. Please edit, for example by changing to “BRG1 deficient cells lines.” Also, please see line 138, as BRG1^{-/-} is also incorrect.

9. How do cells revert to express BRG1? Knowing the mechanism could be insightful.

10. Genetic rescues were not performed in this study.

11. It would be extremely helpful to show the mutation induced by CRISPR.

12. Is the gene editing focused on a functional domain?

- Figure 1 :

- o Strengths: They set the stage for early effects of KD BRG1, provide convincing evidence that BRG1 does in fact slow proliferation by slowing G1/S transition phase

- o Weaknesses: Their control (BRG1 +/+) was not infected with any guide, typically in CRISPR experiments Rosa or some other guide should be used to rule out potential off target effects due to DNA editing by sgRNA. It's odd that in figure (e) the number of cells in G1 doesn't seem significantly different compared to controls, but in (f) they show significance only for clone 2. Additionally, it would be nice to see whether BRG1 is totally knocked out—overexpose and see if there's still a signal

- Figure 2 :

- o Strengths: Confirmed that clone 2 population successfully represses BRG1 throughout the 8-month duration. Provide evidence that fitness in BRG1 KD does increase throughout the experiment when compared to early phase KD cells.

- o Weaknesses: Figure (e) is somewhat confusing-I know they are comparing early vs late and are interested in BRG1 KD, but I expected there to be significant difference for G1 between early WT and early BRG1 ^{-/-}; I expected BRG1 KD cells having significantly more cells in G1.

- Figure 3 :

- o Strengths: Provided various data to drive home the point that BRG1 KD cells have an increase in # of chromosomes at middle and late phase when compared to WT cells.

- o Weaknesses: I strongly feel that there should an explanation as to why gain in chr.18, specifically, confers aneuploidy tolerance. Another concern I have is that they did see a gain in ch21 at all phases (Early, Mid, Late)-but ruled out the possibility that it confers aneuploidy tolerance because (1) the frequency of gain of ch21 remained the same through all 8 months, and (2) clone 1 didn't reproduce this same aneuploidy in ch21. I don't feel either of these reasons are enough to rule out the possibility that gain in ch21 doesn't play a role in conferring aneuploidy tolerance. It's possible that this gain can prime for aneuploidy tolerance, and although clone 1 doesn't cause the chr21 gain, they didn't mention what effect clone 3 had---also I find it concerning that one clonal population (clone 2) can confer and sustain gain in chr 21, whilst another clonal population (clone 1) does not—it raises the question of how on target their sgRNAs are to SMARC4, I would like another explanation of why there are such drastic

differences.

- Figure 4 :

- o Strengths: A strength of the PCA plot is that it seems samples divide by BRG1 expression (PC1) and within BRG1 KD, we see samples divide further between Early, Mid and Late phase (PC2). I appreciated the analysis for parsing out which pathways may be conferring aneuploidy tolerance. From my understanding, it seems they first looked at KD vs WT BRG1, and then took these pathways/processes and analyzed them again between Late phase KD vs Early phase KD BRG1 cells
- o Weaknesses: For Figure b they said that they presented the top 30% most variable proteins based on standard deviation—however, people usually parse out significantly variable proteins based on p-value and not standard deviation. I would inquire about why they choose standard deviation? What would the result be if they based this off of p-value? For figures c and d, although I thought it was an interesting way to go about analyzing pathways etc., I would have liked to have seen a comparison between WT and Late phase. The way it is presented, it drives home the point that things that were once upregulated in Early phase BRG1-deficient cells are now downregulated in Late phase BRG1 KD cells (which would be in agreement with their cell cycle data where late phase KD cells look more like WT than early phase). However, I feel a stronger way to drive this point home would be to look at these same pathways between WT and Late phase BRG1 KD cells. Lastly, in Figure e, I'm interested how they can explain Late phase BRG1 -/- cells having upregulation of proteins associated with G1 phase (similar profile to Early BRG1 KD cells) but yet, the cell cycle analysis shows that Late Phase is comparable to WT (Figure 2d).

- Figure 5:

- o Strengths: In figures c and d, they introduce glycolysis as a pathway that may play a significant role in conferring aneuploidy tolerance. Additionally, they provide preliminary evidence that changes in proteasome-associated protein expression may also confer aneuploidy tolerance (Figure e).
- o Weaknesses: I don't feel Figures a and b provide new data, as we already saw these results in Figure 4c. The author's attempt to associate changes in glycolysis and proteasome related pathways/proteins to conferring aneuploidy tolerance must be strengthened.

- Figure 6:

- o Strengths: None

- o Weaknesses: What is the purpose of having both Figures A and B? I feel Figure C is a reach, as they just found two genes and/or proteins that are dysregulated and have association with immunoproteasome.

- Figure 7:

- o Strengths: I appreciate that they included human data and that they looked at BRG1 in other cancers (b/c & d/e).

- o Weaknesses: I don't feel Figure A is worth placing as a main figure as it seems to me to be a negative result. They make the argument that wt BRG1 cells usually lose Ch.18 but mutated BRG1 cells don't, however there isn't any gain either. Additionally, I'm wary of them comparing their results to that of patient samples where BRG1 is mutated and not necessarily knocked down/out like in their cell lines.

Overall, I feel this paper is an interesting attempt to assess how loss of BRG1 can confer aneuploidy tolerance, and in the broad scheme of things lead to adaptive evolution in cancer. This is a very complicated theory to address and the authors nicely summed it up in their last paragraph of the Discussion session. However, this paper needs to be more focused in order to elucidate pathway(s) that confer aneuploidy tolerance. From my perspective, it seems they knocked out BRG1, looked at the proteome and transcriptome, and cherry-picked processes that could explain aneuploidy tolerance

without further investigation. Furthermore, I am not fully convinced that BRG1 confers aneuploidy tolerance, or that ch.18 plays a major role in this—it would be interesting for the authors to delve into why they believe ch.18 plays such a special role in this and what's its association with BRG1. Additionally, their title claims that loss of BRG1 also leads to improved fitness is misleading, given that this fitness is being compared to Early phase BRG1 $-/-$ and not WT. Lastly, I think it would be interesting to look at the data from clones 1, and 3—although they regain BRG1 expression it would be interesting to see whether all it takes is changes in the Early phase (when BRG1 is KD in all clones), to confer aneuploidy/aneuploidy tolerance—I think of it as an unintentional rescue of BRG1. With that said, I do not feel this work is developed sufficiently for publication, the questions and potential findings when further explored, could end up having a big impact on the field.

Reviewer #3 (Remarks to the Author):

Schiavoni et al report on new studies related to a role for BRG1 in aneuploidy tolerance. Data demonstrate that BRG1 leads to an initial loss of fitness but that continued growth leads to recovery. Further, this recovery was related 2 outcomes – re-expression of the CRISPR inactivated BRG1 or gain of chr19. Studies on the line which gained chr19 indicated that, overtime, this clone recovered growth and extensive MS and genomic analysis indicated that these lines had altered metabolism and metabolic pathway consistent with previous studies on aneuploidy tolerance. Interestingly, these cells also displayed upregulation of proteasome components, as might be expected to deal with increased protein synthesis related to chromosome gain. The authors then go on to demonstrate that BRG1 negative lung adenocarcinomas exhibit increased aneuploidy, providing support for the data generated in their model system. In summary, this is a very interesting paper providing some additional insight into how BRG1 loss can lead to aneuploidy, and detailing some of the metabolic rewiring which is associated with this process.

Points for consideration.

1. 2/3 clones exhibited re-expression of BRG1, indicating strong selection pressure for retaining BRG1. Presumably reversion arises through mutation of the previously CRISPRed BRG1. Does the sequence of BRG1 in the revertants provide any clue as to how function was restored? E.g. frame shift mutations etc.
2. The p53 results are interesting, particularly the increase in p21 levels. Do the elevated p21 levels account for the slow growth of these lines and the development of aneuploidy? If p21 is targeted, do they recover growth? Or undergo apoptosis? Overexpression of cyclin D1 (as the authors note) may oppose p53 function.

Responses to Reviewers

We thank all three reviewers for their very thorough and insightful comments. We agree that a major outstanding question in our previous manuscript was whether the pathway changes that we identified in the BRG1 deficient cells were responsible for altered tolerance to aneuploidy. This turned out to be a very challenging undertaking since the BRG1 deficient cells grow so slowly (and we had pandemic issues, end of a contract, and maternity leave), so we apologise for the slow resubmission process. Nevertheless, we were able to experimentally address this question along with other points raised by the reviewers as outlined below (responses are in blue italics). We now provide evidence that the BRG1 deficient cells are indeed more tolerant of aneuploidy than the parental cells (new Figure 6). We appreciate that the manuscript has benefited from these changes.

Reviewer #1 (Remarks to the Author):

Summary. The authors use Crispr to mutate BRG1 in HCT116 cells and create 3 clones. This suppresses proliferation and over time the clones adapt. Two clones appear to restore Brg1 while a third appears to remain BRG1-deficient, but note that the characterisation here is not sufficiently in depth. The clone that remains BRG1-deficient gains copies of chromosome 18, leading the authors to conclude that BRG1 deficiency enhances aneuploidy tolerance, as explicitly stated in the title. Proteomics evokes hypotheses based on deregulation of hypoxia/glycolysis and immunoproteasome pathways. Testing one of these would make the story considerably more compelling for Nat Comms. More problematic is the lack of evidence to support aneuploidy tolerance. Inhibition of BRG1 imposes a fitness cost exerting a strong selective pressure, evidenced by the two reversion events. Therefore, an alternative explanation is that a random missegregation event yields a subclone with extra copies of 18 which can now restore proliferation in the absence of BRG1 and thus outcompete the original euploid cells. To evoke aneuploidy tolerance, the authors would need to experimentally induce chromosome missegregation to generate random aneuploidies and show that aneuploid cells persist in the BRG1-deficient population for longer.

Critique. Aneuploidy is a hallmark of human cancers despite considerable evidence that its effects are mostly detrimental to cellular and organismal fitness (classically known as the aneuploidy paradox). Therefore, understanding the mechanisms that contribute to aneuploidy's origin, such as chromosome instability (CIN), and its tolerance is of critical interest to the cancer biology community. In their manuscript, Schiavoni & Zuazua-Villar et al. propose that the ablation of BRG1 in the microsatellite unstable, chromosomally stable colon cancer cell line HCT116 leads to aneuploidy tolerance over time. Using CRISPR, the authors generate cell lines deficient for BRG1. Two of the three cell lines regain expression of BRG1 over time thus the authors focus on one clone which remains BRG1 deficient. As these cells are passaged, a gain of chr. 18 is noted. Proteomic analysis reveals an upregulation of glycolysis related proteins and an upregulation of the immunoproteasome (IP) at the early timepoint in the BRG1^{-/-} vs WT cells. The former is consistent with previous observations in trisomic cells (Williams et al. (2008), PMID: 18974345, cited). The latter finding is novel and intriguing since the IP has recently been shown to protect thymic epithelial cells from proteotoxic stress a well-established consequence of aneuploidy (PMID: 29186691, omitted). The authors then proceed to interrogate cancer genomics data stratifying for BRG1 mutation status to evaluate if its mutation correlates with aneuploidy in human cancers which holds true in part. While BRG1 deficiency has been known to cause mitotic abnormalities (Bourgo et al. (2009), PMID: 19458193, cited), replication stress (Cohen et al. (2010), PMID: 20571081, omitted) and activation of the decatenation checkpoint (Dykhuizen et al. (2013), PMID: 23698369, omitted), a role for aneuploidy tolerance has not previously been recognized. Therefore, this manuscript presents an interesting

dataset that will be appreciated by the field.

In terms of strengths, the manuscript is very well written and easy to follow for the most part. Likewise, the data is presented mostly appropriately, and the figures are well prepared. The choice of model system is clear. The characterization of the mitotic phenotype (or rather absence thereof) in the BRG1mut cells deserves to be commended due to the absence of an easily detected difference. The observation that the IP is upregulated in BRG1mut, aneuploid cells is highly interesting in terms of aneuploidy tolerance and indeed novel. The proteomic analysis also corroborates previous mRNA expression analyses that suggested a role for hypoxia and glycolysis in aneuploidy tolerance. In terms of weaknesses, the characterization of the BRG1mut cells is insufficient, especially in the absence of the expected mitotic aberrations observed in murine BRG1-deficient models. Aneuploidy tolerance is a strong term that is overstated in this context as the evidence presented only justifies saying that BRG1 deficiency creates a selective pressure that is buffered by a gain of chr. 18. If this is the new angle, it could be tested by mutating BRG1 in already trisomic HCT116 cells (Vasudevan et al, (2020), PMID: 32097652) but this is probably beyond the scope of this manuscript.

Major issues before publication in Nat Comms can be recommended:

1. The clones need to be better characterised with respect to the nature of the BRG1 mutations and reversions.

This is an important point, and has now been done. First, we sequenced the BRG1 KO clones and identified the mutations present in the cell lines upon generation (new Supplementary Figure 1). Clone 2, which we used for the majority of experiments in the original version of the manuscript, has a 1bp deletion in both alleles of exon 2 leading to a frameshift.

In addition, we sequenced the two KO clones at late time points when BRG1 re-expression is evident. Clone 3, which had a 1bp deletion and a 2 bp insertion in the two alleles upon generation, yielded a single wt BRG1 sequence at late time points. This could arise through a combination of reversion and recombination events or through outgrowth of contaminating wt cells. Since we couldn't rule out contamination, we dropped clone 3 from the revised manuscript.

Clone 1 had a 1bp del and 2bp insertion in the two BRG1 alleles upon generation (new Supplementary Fig 1). At late time points, sequencing yielded a mix of products for Clone 1. We therefore cloned PCR products spanning the guide RNA binding site and sequenced these to understand what changes were present. We identified three independent products; the original 1bp deletion and 2bp insertion, and we also found a 3bp deletion, which would lead to the loss of a single codon and restoration of BRG1 expression (new Supplementary Fig. 2a).

In the revised manuscript, we use clones 1 and 2 at early time points (when both are BRG1-deficient) to interrogate tolerance to aneuploidy (new Figure 6).

2. Evidence of bona fide aneuploidy tolerance needs to be provided, or the emphasis of the story is changed to one that focuses on gain of chromosome 18 being able to buffer loss of BRG1 function.

This is a key point, thank you. Our data in the previous manuscript were consistent with elevated tolerance to aneuploidy, but in the revised manuscript, we directly tested this. Specifically, we treated cells with Mps1 inhibitors to induce chromosome missegregation in the isogenic cell lines. We find that both BRG1 knockout clones (at early time points – prior to any fitness or chromosome gains) survive proportionately better than the BRG1 proficient parental cells (new Figure 6).

Moreover, we analysed the karyotypes of the surviving cells and find that there are more aneuploid survivors in the BRG1 KO cells than in the BRG1 proficient parental cells, consistent with increased tolerance. Importantly, we have done these experiments in two independent KO clones (clones 1 and 2). These data provide direct evidence of greater tolerance to aneuploidy in the absence of BRG1.

3. The authors need to test whether upregulation of hypoxia/glycolysis and/or immunoproteasome pathways is responsible for fitness restoration in the absence of BRG1.

Upregulation of these pathways precedes fitness restoration, suggesting that these contribute to aneuploidy tolerance, but not increased fitness. We have added text to clarify this in the revised manuscript.

4. Weave in additional relevant citations highlighted above.

This has been done, thanks for the suggestions.

Other issues to address/consider:

Line 72, the mouse ES cell data needs to be rationalised with the new data in the Discussion.

Done, thank you.

Line 107, explicitly state that by depletion you are referring to an RNAi experiment.

Done, thank you.

Line 107, call out of supplementary figures 1c-e precedes supplementary figures 1a-b.

The text has been modified to put these in order.

Line 113, FACS profiles too harshly gated to support statement.

We provided additional FACS profiles to support the statement (new Supplementary Fig 1h).

Line 115, 1e and f somewhat redundant.

We appreciate this is true, but think providing the data in both formats has value.

Line 124, need to demonstrate that clone 2 does indeed remain “negative”. See major issue above, what is the nature of the initial BRG1 mutation in the three lines? In 1 and 3, it is a simple single nucleotide deletion that reverts back to WT? Does clone 2 have a more complex indel that is harder to revert, explaining why it remains fixed? Or does clone 2 also restore partial BRG1 function via a different mechanism? E.g. a splice variant not detected by the antibody used. Can the mass spec data be mined/interpreted to rule this out? Note the recent issues with BUB1. It is also important to rule out the trivial explanation of outgrowth of wildtype cells due to the original lines not being clonal. I suspect this is not the case otherwise they would have outgrown prior to the initial characterisations but this needs to be formally addressed in the text.

Yes, this is an important point, and we provide additional information in the revised manuscript to address this. The sequencing data are provided as described above. In addition, we were able to mine the mass spec data to confirm that clone 2 does indeed remain negative throughout the evolution in culture. We re-processed the raw mass spectrometry files using a protein fasta file containing the sequences of 29 potential isoforms of BRG1 (SMARCA4) retrieved from Uniprot. The quantification profiles of all peptides identified do not show expression of BRG1 in the knock-out cells (new Supplementary Fig. 2c). 16 of these peptides map to at least 20 isoform sequences. For some peptides we only observe a minimal mass spec signal (<10% of the WT abundance on average) in the KO cells, most likely due to background co-isolation interference in the isobaric labelling experiment rather than protein expression.

Line 128, 2c would benefit from the comparisons being on the same 6 well plate and quantification.

We appreciate this point but have provided proliferation assays as well, so we prioritised other experiments for the revision.

Line 132, 2e not compelling, are the n=3 biological replicates?

Yes, these data are from 3 biological replicates, and in the revised version we provided more information on gating and replicates.

Line 142, panel 3b can you show the actual chromosome numbers rather than gain or loss, i.e. in any given line is it a gain or loss of 1, 2, 3 or more chromosomes? 3f is better. Can you also correlate chr gains and chr 18 gain in each metaphase? It seems FISH was performed on M-phase spreads already.

Provided in new Supplementary Figure 3a.

Line 152, the array CGH data is nice but as presented doesn't really do the data justice. The authors should stick to array CGH as conventional CGH is something different.

Thanks, we have revised the text to make this more clear.

Line 166, enhance FISH IF to better display the chromosomes.

Done, thanks.

Line 181, the PCA shows very nicely the evolution of the cell line.

Thank you.

Line 189, mass spec nicely corroborated westerns showing loss of BRG1 at all timepoints.

Thanks, and we have further analysed these data to investigate BRG1 levels over time as described above (new Figure S2c).

Line 208, panels 4c and 4d could be annotated with titles to better display what is being compared.

Good suggestion, this has been done, thanks.

Line 211, 4d, 5a and 5c are missing statistical analysis of scaled protein abundance.

The statistics have been added, thank you.

Line 217, the language is getting a bit sloppy: "When we look more closely at regulators of the G1/S transition, we find that ..."

The text has been revised to improve clarity.

Line 241, good effort to rule out mitotic errors.

Thanks.

Line 248, need to formally test whether it is aneuploidy tolerance, i.e. put BRG1 mutants through an Mps1 inhibitor washout protocol to induce random aneuploidies and show that they are not as rapidly eliminated compared to wildtype. See major issue above.

Excellent suggestion, thank you, now added as new Figure 6. We used two different Mps1 inhibitors to induce chromosome missegregation and analysed growth over time across a range of doses. We found the BRG1 KO cells are able to survive better than the parental cells after Mps1 inhibition. We then examined the karyotypes of the survivors and found that there were more aneuploid cells in the BRG1 KO surviving cells post-treatment, consistent with increased survival reflecting a greater tolerance of aneuploidy. Finally, we used two independently derived BRG1 KO clones (clones 1 and 2 at early time points) for these new experiments in order to mitigate against clone-specific effects.

Line 253, original work should be cited at least alongside review Thompson & Compton (2010), PMID: 20123995.

This has been added.

Line 262, reference 18 refers to Cyclin D overexpression in tolerance of whole genome duplication in diploid cells, response to tetraploidy might differ from a response to aneuploidy.

We have altered the text to make this clear.

Line 267, why not first point out that hypoxia emerges from the mass spec, then note Hif1 signalling was also recently implicated in aneuploidy tolerance?

We have taken this suggestion, thank you.

Line 278, but none of these ideas are tested, would this not be required for a Nat Comms paper? See major issue above.

Now tested as described above (new Figure 6).

Line 308, not obvious how mRNA levels were analysed, would be helpful to clarify in the text.

We have altered the text to make this clear.

Line 311, PSM genes are upregulated in clones 1 and 2, this is confusing as clone 1 restores BRG1 function. Was this done at an early time point before reversion?

Yes, this experiment was done at an early time point before reversion. We have altered the text to make this clearer.

Line 330, as above the authors present a hypothesis but do not test it. Again, see major issue above.

Now tested as described above (new Figure 6).

Line 349 should read "did not".

This has been changed.

Line 478 should read “manufacturer”.

This has been changed.

Line 547, bioinformatics analysis of mass spec experiment is missing.

This has been added, apologies for the oversight.

Reviewer #2 (Remarks to the Author):

The authors had previously found that loss of the BAF180 subunit that is specific to the PBAF complex in which BRG1 is the catalytic subunit in U2OS cells leads to chromosome instability and enhanced aneuploidy. Importantly, they found the ES cells lacking Baf180 were aneuploid with two recurring chromosome gains, but had no observable compromise in fitness. These findings suggest that the PBAF complex regulates aneuploidy, and that aneuploid cells can ‘adjust’ genomically to attain an adequate level of fitness.

Synopsis : Schiavoni et.al’s main focus in this paper was exploring what proteins and pathways are dysregulated when BRG1 is knocked down in the context of HCT116 cell line (colorectal cancer cell line); Schiavoni et.al ultimately concluded that the loss of BRG1 and subsequent dysregulation of processes associated with BRG1 expression, allows for tolerance of aneuploidy over 8 months in the HCT116 cell line. To study BRG1’s association to aneuploidy tolerance, Schiavoni et.al used CRISPR to inactivate SMARCA4 (encodes BRG1) in HCT116 and created 3 different clonal cell lines with 3 individually sgRNAs in hope of inactivating SMARCA4 and therefore knocking down BRG1. They then monitored these cell lines in-vitro for 8 months, and analyzed the effects of BRG1 at three time points: early, middle (4 months), and late (8 months).

Schiavoni et.al successfully showed that when BRG1 is knocked down: (1) it causes a decrease in rate of proliferation compared to controls which is supported by, (2) a cell count assay, and flow cytometry assay showing that BRG1 KD cells move more slowly in the G1 phase compared to control cells. To ultimately test the effect of BRG1 KD on aneuploidy, they (3) used metaphase FISH with a probe against chromosome 18, and showed that BRG1 KD cells in middle and late phases had a gain in chromosome 18 compared to controls (Figure 3d). Interestingly, they saw that late phase cells with BRG1 KD slowly regained fitness, which was assessed through proliferation rate (cell count) and cell cycle analysis using flow cytometry, which showed that late phase BRG1 KD cells had similar cell cycle profile to that of the control cells.

To determine whether BRG1 causes aneuploidy vs sets the stage for aneuploidy, they assessed chromosomal instability via quantifying micronuclei during cell division, and found no significant difference in the number of micronuclei during division between KD and WT BRG1 in early, middle, or late phase. From these results, they concluded that loss of BRG1 does not directly cause aneuploidy, but rather, may set the stage for aneuploidy tolerance. They coupled these results to proteome and RNA_seq analysis where they first (1), compared WT to early phase KD cells, and then analyzed these same pathways and proteins that were deregulated in early phase and (2) compared early phase and late phase BRG1 KD cells. From their results, Schiavoni et.al postulate that some of these processes (i.e. increase in glycolysis and hypoxia associated proteins and pathways) may confer aneuploidy tolerance. The culmination of this work gave life to their title that loss of BRG1 and its subsequent dysregulated processes may confer aneuploidy tolerance which could potentially set the stage for oncogenesis.

Concerns:

1. The initial work using Baf180 deficient ES cells was based on strong data since it used normal cells; here the work is much less convincing since it is based on a single human colorectal cancer cell line. *We appreciate the limitations of working in a single cell line, but have also explored the cancer databases and the data from these suggest that the effects of BRG1 deficiency on aneuploidy and on upregulation of the immunoproteasome is not limited to this cell line. We are currently exploring other model systems to interrogate the impact of SWI/SNF activity on these pathways and tolerance of aneuploidy.*

2. BRG1 is present in multiple complexes, so it is not at all clear that is working through PBAF, as appears to be assumed here.

This is a good point. This study was inspired by our observations from cell lines lacking the PBAF specific subunit PBRM1/BAF180, but our data is specifically focused on BRG1, which is indeed present in multiple complexes. We have modified the text to highlight this.

3. The authors discuss apoptosis, and do not observe it upon BRG1 depletion. Was cellular senescence considered?

Yes, we looked for evidence of senescence upon BRG1 depletion, but find no evidence for it. We've stated this in the revised manuscript.

4. While it was great that three subclones were initially established, two of these clones regained BRG1 expression. Therefore, the work is based on only a single subclone.

In the revised manuscript, we have expanded the work to go beyond study of a single subclone. Specifically, we directly test whether the loss of BRG1 increases tolerance to aneuploidy. Since the effects on pathways related to aneuploidy tolerance are evident immediately upon loss of BRG1, we were able to use two independently derived BRG1 KO clones by testing them at early passage prior to fitness restoration or BRG1 re-expression. In addition, we used siRNA where possible to look at the effects of BRG1 loss through an orthogonal approach.

5. It is not clear whether gain of chromosome 18 was one of the two chromosome gains found in BAF180 deficient U2OS cells.

We did not identify the chromosome changes associated with BAF180 loss in U2OS cells. Our prediction is that chromosome alterations will be cell type specific (and even culture condition specific). We are currently testing this possibility.

6. When discussing the mouse ES cells lacking Baf180, it would be preferable to use "ES" cells rather than "MES."

Thanks, we have changed this in the revised manuscript.

7. Please use Baf180 instead of BAF180 to designate mouse.

Thanks, we have changed this in the revised manuscript.

8. Referring to the "BRG1 null cell lines" (for example line 111) is incorrect, as "null" refers to alleles, but here you are discussing the protein. Please edit, for example by changing to "BRG1 deficient cells lines." Also, please see line 138, as BRG1^{-/-} is also incorrect.

Thanks, we have changed this in the revised manuscript.

9. How do cells revert to express BRG1? Knowing the mechanism could be insightful.

As described in the responses to Reviewer 1, we have now sequenced the cells with BRG1 re-expression. In clone 1 that has re-expression at late time points, we find a new sequence with a 3bp deletion that restores in-frame translation of the protein. The deleted 3bp encompasses the 1bp

deletion in the original clone, suggesting a mechanism for reversion. We have added the sequencing information to the revised manuscript (Supplementary Fig. 1a and 2a).

10. Genetic rescues were not performed in this study.

We made considerable efforts to perform rescue experiments, but transient transfections were too inefficient, and we were unable to recover stable cells harbouring BRG1 expression constructs. This is likely due to the poor fitness of the BRG1 knockout cells.

11. It would be extremely helpful to show the mutation induced by CRISPR.

We sequenced all the clones and provide sequencing information and the impact on the coding sequence in the revised manuscript (new Supplementary Figure 1).

12. Is the gene editing focused on a functional domain?

The guide RNA was directed to exon 2 of the SMARCA4 (BRG1 encoding) gene at the start of the protein coding sequence. The impact of the gene editing changes on the coding sequence of BRG1 is provided in the revised manuscript (new Supplementary Fig. 1a).

- Figure 1 :

- o Strengths: They set the stage for early effects of KD BRG1, provide convincing evidence that BRG1 does in fact slow proliferation by slowing G1/S transition phase

- o Weaknesses: Their control (BRG1 +/-) was not infected with any guide, typically in CRISPR experiments Rosa or some other guide should be used to rule out potential off target effects due to DNA editing by sgRNA. It's odd that in figure (e) the number of cells in G1 doesn't seem significantly different compared to controls, but in (f) they show significance only for clone 2. Additionally, it would be nice to see whether BRG1 is totally knocked out—overexpose and see if there's still a signal

These are all good points. In response to each:

We used siRNA depletion (with multiple different siRNAs) and find that the phenotypes are similar to the KO cells, suggesting that these phenotypes are not a consequence of off-target effects.

The number of cells in G1 phase in both clones is different compared to the controls in 1e.

We now include data for clone 1 in 1f, which is also significant. Note that we dropped the use of clone 3 in the revised manuscript as described above in the responses to Reviewer 1.

As described above, we further characterised the KO clones so we know they have frameshift mutations in both alleles and we mined the mass spec data for clone 2 to ensure no residual or partial protein expression.

- Figure 2 :

- o Strengths: Confirmed that clone 2 population successfully represses BRG1 throughout the 8-month duration. Provide evidence that fitness in BRG1 KD does increase throughout the experiment when compared to early phase KD cells.

- o Weaknesses: Figure (e) is somewhat confusing-I know they are comparing early vs late and are interested in BRG1 KD, but I expected there to be significant difference for G1 between early WT and early BRG1 -/-; I expected BRG1 KD cells having significantly more cells in G1.

This is a good point and we have added additional gating information, replicate information and analysis to the revised figure to make this more clear. The early BRG1 KO cells do have proportionately more G1 phase cells than the parental cells.

- Figure 3 :

- o Strengths: Provided various data to drive home the point that BRG1 KD cells have an increase in # of chromosomes at middle and late phase when compared to WT cells.

- o Weaknesses: I strongly feel that there should an explanation as to why gain in chr.18, specifically,

confers aneuploidy tolerance. Another concern I have is that they did see a gain in ch21 at all phases (Early, Mid, Late)-but ruled out the possibility that it confers aneuploidy tolerance because (1) the frequency of gain of ch21 remained the same through all 8 months, and (2) clone 1 didn't reproduce this same aneuploidy in ch21. I don't feel either of these reasons are enough to rule out the possibility that gain in ch21 doesn't play a role in conferring aneuploidy tolerance. It's possible that this gain can prime for aneuploidy tolerance, and although clone 1 doesn't cause the chr21 gain, they didn't mention what effect clone 3 had---also I find it concerning that one clonal population (clone 2) can confer and sustain gain in chr 21, whilst another clonal population (clone 1) does not---it raises the question of how on target their sgRNAs are to SMARC4, I would like another explanation of why there are such drastic differences.

Apologies for the lack of clarity in the previous manuscript. The model is that the tolerance to aneuploidy is conferred by the changes in the immunoproteasome and hypoxia/glycolysis pathways - prior to any gain of chromosomes. In the revised manuscript, we laid out this model in more detail and directly tested it by looking at survival after inducing chromosome missegregation (with Mps1 inhibitors) and at the karyotypes of the surviving cells. We used early passage cell lines (both clones 1 and 2) prior to any fitness changes or chromosome gains, and consistent with this model, we found that the BRG1 KO cells are able to survive chromosome missegregation proportionately better than the parental cells, and more of the survivors are aneuploid. These data are in new Figure 6.

- Figure 4 :

- o Strengths: A strength of the PCA plot is that it seems samples divide by BRG1 expression (PC1) and within BRG1 KD, we see samples divide further between Early, Mid and Late phase (PC2). I appreciated the analysis for parsing out which pathways may be conferring aneuploidy tolerance. From my understanding, it seems they first looked at KD vs WT BRG1, and then took these pathways/processes and analyzed them again between Late phase KD vs Early phase KD BRG1 cells *This is correct - we performed the pathway/process analysis independently for KO vs WT BRG1 and then between late time point KO vs early time point KO cells and we looked for common and unique pathways/processes. We have added titles to the panels in Figure 4 to improve clarity.*

- o Weaknesses: For Figure b they said that they presented the top 30% most variable proteins based on standard deviation---however, people usually parse out significantly variable proteins based on p-value and not standard deviation. I would inquire about why they choose standard deviation? What would the result be if they based this off of p-value? For figures c and d, although I thought it was an interesting way to go about analyzing pathways etc., I would have liked to have seen a comparison between WT and Late phase. The way it is presented, it drives home the point that things that were once upregulated in Early phase BRG1-deficient cells are now downregulated in Late phase BRG1 KD cells (which would be in agreement with their cell cycle data where late phase KD cells look more like WT than early phase). However, I feel a stronger way to drive this point home would be to look at these same pathways between WT and Late phase BRG1 KD cells. Lastly, in Figure e, I'm interested

how they can explain Late phase BRG1 -/- cells having upregulation of proteins associated with G1 phase (similar profile to Early BRG1 KD cells) but yet, the cell cycle analysis shows that Late Phase is comparable to WT (Figure 2d).

Thanks for these comments. First, the proteins shown in Figure 4b have been selected based on ANOVA test and Benj. Hoch. FDR<0.05 (adjusted p-value). However, there are many proteins with Benj. Hoch. FDR<0.05 but with small differences between the sample groups that may not be biologically relevant. We therefore used the standard deviation as an additional filter on top of adjusted p-value to select the most divergent proteins between the groups and to exclude proteins with very small differences.

Second, thanks, this was a good suggestion to compare the parental cells to the KO cells at the late time point. We find that the KO at late time points still has misregulation of hypoxia, but no longer

shows substantial misregulation of DNA replication or G1/S transition pathways, consistent with the phenotypic changes. These data are provided in new Supplementary Figure 3g.

Finally, the misregulation of G1/S phase regulatory proteins is an interesting point. The changes at early time points in the KO cells are consistent with their altered cell cycle profiles. However, we agree that at late time points, when cell cycle profiles are similar to the parental cell line, the prediction would be that the G1/S protein levels in the KO would also be similar to the parental cell line. While this is true for a subset of proteins in this analysis, the striking overexpression of CDKN1A, CCND1, CCND3, and CDK6 does not track with the changes in cell cycle profiles. We speculate that these proteins have been deregulated at a post-translational level in the KO cells at late time points.

- Figure 5:

- o Strengths: In figures c and d, they introduce glycolysis as a pathway that may play a significant role in conferring aneuploidy tolerance. Additionally, they provide preliminary evidence that changes in proteasome-associated protein expression may also confer aneuploidy tolerance (Figure e).

- o Weaknesses: I don't feel Figures a and b provide new data, as we already saw these results in Figure 4c. The author's attempt to associate changes in glycolysis and proteasome related pathways/proteins to conferring aneuploidy tolerance must be strengthened.

This is a good point, thanks. In the revised manuscript, we directly test the tolerance of aneuploidy in the KO cells, and these data are provided in new Figure 6.

- Figure 6:

- o Strengths: None

- o Weaknesses: What is the purpose of having both Figures A and B? I feel Figure C is a reach, as they just found two genes and/or proteins that are dysregulated and have association with immunoproteasome.

Panel B (from the original manuscript) was an analysis of mRNA levels to determine whether the protein level changes (from original Fig. 6A) were arising through transcriptional or post-transcriptional mechanisms. The STAT1 and STAT2 proteins in original Fig. 6C are key regulators of the pathway. We have now merged the data on the immunoproteasome with the original Figure 5 and moved the analysis of STAT1 and STAT2 to Supplementary Fig. 6. The new Figure 6 provides evidence of aneuploidy tolerance upon BRG1 loss, which we appreciate strengthens the manuscript.

- Figure 7:

- o Strengths: I appreciate that they included human data and that they looked at BRG1 in other cancers (b/c & d/e).

- o Weaknesses: I don't feel Figure A is worth placing as a main figure as it seems to me to be a negative result. They make the argument that wt BRG1 cells usually lose Ch.18 but mutated BRG1 cells don't, however there isn't any gain either. Additionally, I'm wary of them comparing their results to that of patient samples where BRG1 is mutated and not necessarily knocked down/out like in their cell lines.

Our interpretation of the data from original Fig. 7a is that there is a relationship between genes on chr18 and the fitness of BRG1 deficient cells. However, we appreciate that it is unrelated to the main point of the manuscript (tolerance of aneuploidy), so we moved it into Supplementary Fig. 7a. We also make the point in the revised manuscript that many (if not most) cancer-associated mutations of BRG1 are loss of function mutations, many of which will yield no detectable protein, which makes comparison with the CRISPR-mediated KO more physiologically relevant.

Overall, I feel this paper is an interesting attempt to assess how loss of BRG1 can confer aneuploidy tolerance, and in the broad scheme of things lead to adaptive evolution in cancer. This is a very complicated theory to address and the authors nicely summed it up in their last paragraph of the Discussion session. However, this paper needs to be more focused in order to elucidate pathway(s) that confer aneuploidy tolerance. From my perspective, it seems they knocked out BRG1, looked at the proteome and transcriptome, and cherry-picked processes that could explain aneuploidy

tolerance without further investigation. Furthermore, I am not fully convinced that BRG1 confers aneuploidy tolerance, or that ch.18 plays a major role in this—it would be interesting for the authors to delve into why they believe ch.18 plays such a special role in this and what's its association with BRG1. Additionally, their title claims that loss of BRG1 also leads to improved fitness is misleading, given that this fitness is being compared to Early phase BRG1 $-/-$ and not WT. Lastly, I think it would be interesting to look at the data from clones 1, and 3—although they regain BRG1 expression it would be interesting to see whether all it takes is changes in the Early phase (when BRG1 is KD in all clones), to confer aneuploidy/aneuploidy tolerance—I think of it as an unintentional rescue of BRG1. With that said, I do not feel this work is developed sufficiently for publication, the questions and potential findings when further explored, could end up having a big impact on the field.

We have changed the title to reflect the fact that it is a recovery of fitness, not improved fitness.

Thank you for highlighting that.

In response to the final point, as outlined above, the model is that early changes upon BRG1 loss lead to aneuploidy tolerance and we directly tested this in the revised manuscript.

Reviewer #3 (Remarks to the Author):

Schiavoni et al report on new studies related to a role for BRG1 in aneuploidy tolerance. Data demonstrate that BRG1 leads to an initial loss of fitness but that continued growth leads to recovery. Further, this recovery was related 2 outcomes – re-expression of the CRISPR inactivated BRG1 or gain of chr19. Studies on the line which gained chr19 indicated that, overtime, this clone recovered growth and extensive MS and genomic analysis indicated that these lines had altered metabolism and metabolic pathway consistent with previous studies on aneuploidy tolerance. Interestingly, these cells also displayed upregulation of proteasome components, as might be expected to deal with increased protein synthesis related to chromosome gain. The authors then go on to demonstrate that BRG1 negative lung adenocarcinomas exhibit increased aneuploidy, providing support for the data generated in their model system. In summary, this is a very interesting paper providing some additional insight into how BRG1 loss can lead to aneuploidy, and detailing some of the metabolic rewiring which is associated with this process.

Points for consideration.

1. 2/3 clones exhibited re-expression of BRG1, indicating strong selection pressure for retaining BRG1. Presumably reversion arises through mutation of the previously CRISPRed BRG1. Does the sequence of BRG1 in the revertants provide any clue as to how function was restored? E.g. frame shift mutations etc.

This is an interesting point, and it does indicate strong selection pressure for BRG1 function. As described above, we have now sequenced the clones at early and late time points. In clone 1 that has re-expression at late time points, we find a new sequence with a 3bp deletion that restores in-frame translation of the protein. The deleted 3bp encompasses the 1bp deletion in the original clone, suggesting a mechanism for reversion. The sequencing data are provided in the revised manuscript (Supplementary Fig. 1a and 2b).

2. The p53 results are interesting, particularly the increase in p21 levels. Do the elevated p21 levels account for the slow growth of these lines and the development of aneuploidy? If p21 is targeted, do they recover growth? Or undergo apoptosis? Overexpression of cyclin D1 (as the authors note) may oppose p53 function.

We agree that this is an interesting change, so we tested the effect of p21 depletion on the growth of the KO cells. We find that p21 depletion leads to a decrease in proliferation rate in both the KO and

the parental cells, so we can conclude that the elevated p21 levels do not account for the slow growth of the KO cells. It is not clear what the significance of the elevated p21 levels is, but it is possible that the combination of changes in protein levels is important, and we are currently looking at this in more detail in other systems.

REVIEWERS' COMMENTS

Reviewer #1 (Remarks to the Author):

Despite a number of challenges, the authors have submitted a revised manuscript that is greatly improved. In a very detailed response they have addressed the 4 major issues. In particular, they have characterised the Crispr mutations, and the Mps1 inhibitor experiment addresses the tolerance issue. In addition they have addressed all the other issues in what was in retrospect a very thorough initial review. Consequently the manuscript is greatly improved and the authors are to be congratulated for their efforts.

Reviewer #2 (Remarks to the Author):

The manuscript by Shiavoni et al entitled "Aneuploidy tolerance caused by BRG1 loss allows chromosome gains and recovery of fitness" has been extensively revised, and the major concerns of the previous submission have adequately addressed.

Please note: Line 70 should be written "Baf180^{-/-}" to represent the mouse genotype. Please also check line 457 to ensure that the nomenclature "BRG8" refers to human cells, as written.

Reviewer #3 (Remarks to the Author):

This is an excellent paper showing a key role for BRG1 in aneuploidy. The authors have provided strong new data (e.g. figure 6) to support their conclusions, and have extensively addressed the many comments raised. Overall, this is a novel paper providing new insight into BRG1 function and how tumor cells may tolerate aneuploidy. No additional changes are needed.

Responses to Reviewers

Aneuploidy tolerance caused by BRG1 loss allows chromosome gains and recovery of fitness

Schiavoni and Zuazua-Villar, et al

NCOMMS-20-13217A

Thanks very much for the positive feedback and comments, they were very much appreciated by all the authors.

Reviewer #1 (Remarks to the Author):

Despite a number of challenges, the authors have submitted a revised manuscript that is greatly improved. In a very detailed response they have addressed the 4 major issues. In particular, they have characterised the Crispr mutations, and the Mps1 inhibitor experiment addresses the tolerance issue. In addition they have addressed all the other issues in what was in retrospect a very thorough initial review. Consequently the manuscript is greatly improved and the authors are to be congratulated for their efforts.

Reviewer #2 (Remarks to the Author):

The manuscript by Shiavoni et al entitled “Aneuploidy tolerance caused by BRG1 loss allows chromosome gains and recovery of fitness” has been extensively revised, and the major concerns of the previous submission have adequately addressed.

Please note: Line 70 should be written “Baf180^{-/-}” to represent the mouse genotype. Please also check line 457 to ensure that the nomenclature “BRG8” refers to human cells, as written.

Thank you for the feedback. We have modified the text to keep the conventions for mouse and human genes and proteins consistent. We italicised the gene name in Line 70, and we clarified the wording in line 457, which was referring to human cells.

Reviewer #3 (Remarks to the Author):

This is an excellent paper showing a key role for BRG1 in aneuploidy. The authors have provided strong new data (e.g. figure 6) to support their conclusions, and have extensively addressed the many comments raised. Overall, this is a novel paper providing new insight into BRG1 function and how tumor cells may tolerate aneuploidy. No additional changes are needed.